# Genetic and environmental influences on data missingness in developmental cognitive neuroscience
G. Bussu [1] ✉, A. M. Portugal [1], C. Viktorsson [1], I. Hardiansyah[1,2] & T. Falck-Ytter [1,2] ✉

Missing data are common in social and clinical sciences and understanding the causes and patterns of missing data is important for selecting analysis approach and for the interpretation of the remaining data. Yet, knowledge about the factors influencing data loss is limited. Here, we assessed the contribution of genes and environments to data missingness across three experiments of infant brain and behavioural development. The sample consisted of 594 infant twins (330 monozygotic, 152 female, 178 male infants; 264 dizygotic, 132 female, 132 male infants) who were assessed with electroencephalography (EEG), pupillometry, and gaze tracking technologies at 5 months of age. Substantial familial factors (additive genetics and/or shared environment) for data missingness were found across all experiments. The amount of missing data showed only a low correlation across the experiments, suggesting a high degree of specificity in the factors contributing to missingness. The results underscore the need to adopt and improve procedural and analytical strategies that minimise data loss and its negative impacts on study conclusions.

Missing data is a major issue in many disciplines, particularly when working with multivariate data[1,2]. Although the reasons for missingness dictate how much of a problem it represents for statistical analyses and interpretations, the underlying sources of missingness are often not well understood[3–5]. Traditional approaches to deal with missing data in social and clinical science, like listwise deletion, operate under the assumption of missing completely at random (MCAR) data patterns, meaning that the distribution of missing data does not depend on observed variables or non-observed missing values[5]. This is often unrealistic in clinical and social sciences[6]. If this assumption is violated, those methods would introduce a bias in estimated statistics and provide invalid statistical inferences and reduced generalizability[5].

While there is increasing awareness of the statistical complications posed by missing data, it is commonly treated as mere noise[2]. Traditionally, researchers investigate missing data patterns by testing for significant differences between complete cases and those with missing data[7]. This approach provides some insight into the factors associated with data missingness, and previous work of this type has demonstrated how both participant characteristics and study design can influence data loss[8,9]. However, much of the research on missing data has primarily focused on compensating for data loss and optimizing data quality for subsequent statistical analysis, rather than exploring the underlying causes of

missingness. Consequently, we still have limited knowledge about the general factors that influence data loss and the potential value of missing data as a meaningful signal rather than a nuisance. These considerations are particularly relevant in infancy research and clinical studies, where missing data is a prevalent challenge[9–14]. Such missingness can stem from difficulties in maintaining participants' attention and following instructions, equipment limitations, and often small—and in the case of neurodevelopmental conditions, heterogeneous—samples. Beyond complete exclusion from a study due to missing data, participants may also contribute varying numbers of analysable trials, further complicating data interpretation[13].

In this study, we aimed to understand factors influencing missing data patterns in developmental cognitive neuroscience, a discipline in which missing data is highly prevalent. We used a multi-pronged approach, testing influences on two main levels of analysis: experiment-level data missingness (i.e., complete data loss from an experiment) and trial-level data availability (i.e., how many individual trials in each experiment were deemed valid and usable for analysis). By combining the experimental approach with a classical twin design, we investigated the influence of general etiological factors on inter-individual variability in experiment-level data missingness and trial-level data availability across three different experimental methods commonly used in developmental infancy research.

[1]Development and Neurodiversity Lab, Department of Psychology, Uppsala University, Uppsala, Sweden. [2]Center of Neurodevelopmental Disorders (KIND), Centre for Psychiatry Research, Department of Women's and Children's Health, Karolinska Institutet & Stockholm Health Care Services, Region Stockholm, Stockholm, Sweden. ✉e-mail: giorgia.bussu@gmail.com; terje.falck-ytter@psyk.uu.se

In the classical twin design, by comparing correlations between twins within monozygotic (MZ) twin pairs (who share 100% of the segregating DNA) and dizygotic (DZ) twin pairs (who share on average 50% of the segregating DNA), variation in the observed trait can be decomposed into influences from additive genetics (A), shared environment (C), and unique environment (E)[15]. We used twin analysis on measures of missing data both at the experiment level and at the trial level, across three experiments reflecting different areas of developmental cognitive neuroscience (pupillometry[16], gaze tracking[17], and EEG[18]) previously reported in peer-reviewed scientific journals (see illustrations of experimental set up and primary data types in Fig. 1a, b, e, f, i, j). Although the overarching twin study from which these data were taken includes many other behavioural biological measures and data types as well[19], we opted here to focus on *experimental* methods specifically.

While the selected pupillometry, gaze tracking and EEG experiments differ in ways that may lead to distinct patterns of missingness—such as hair quality, scalp size, and sensory sensitivity in EEG[20], or eye physiology in eye tracking[21]—they also share several overarching factors likely to influence all datasets. These include attentional and temperamental traits (e.g., activity level), as well as broader family-related or testing day-related influences.

Against this backdrop, we considered the three experiments to provide a compelling and complementary set of data types for addressing the issue of missingness from a behavioural genetics perspective. Specifically, we examined which etiological factors influence individual variability in the incidence of missing data across different experimental paradigms traditionally used in developmental cognitive neuroscience. This aim and the analytic approach were preregistered[22]; however, as noted in the Methods section, there were deviations in the specific tests conducted.

## Methods

### Preregistration

The current report followed the general aim as preregistered (OSF.IO/RQWVC). Specific hypotheses and analyses were dropped as the result of the review process, as follows: 1) We excluded one hypothesis (preregistration hypothesis 1) which reflected a questionable assumption about the relationship between MCAR and the results of behavioural genetic studies. 2) Due to the low cross-experiment phenotypic correlations and the corresponding non-significant cross-twin-cross-trait correlations, it was not meaningful to conduct formal multivariate behaviour genetic modelling (preregistration hypothesis 2). Consequently, the current report addresses a more general question about the etiological factors influencing data missingness in infancy than was initially planned.

### Participants

Participants in the study were recruited for the BabyTwins Study Sweden (BATSS[9]) in the greater Stockholm area from the Swedish population registry, for a total of 622 same-sex twins (311 pairs). Sex was assigned based on information in the registry.assZygosity was estimated based on DNA sampled from saliva (see ref. 19). All same-sex twins in the population register within the region of interest were contacted, as further described in ref. 19. Ultimately, around 30% of the target population were included in the BATSS study. Ethical approval for the study was granted by the Regional Ethics Board in Stockholm. Prior to participation, written informed consent was obtained from each infant's legal guardians.

The general aim of the BATSS study was to advance understanding of the genetic and environmental contributions to infant development, with a particular focus on brain, attentional, and behavioral domains[19]. By employing a multimethod, deep-phenotyping approach starting at five months of age with follow-up until age three years, the study addressed limitations of previous twin research—such as late onset of data collection, overreliance on questionnaire or behavioural data, and small sample sizes.

General inclusion criteria for the BATSS study ensured that the infants i) were part of same-sex twin pairs that lived together, ii) were aged between 5 and 6 months, iii) had at least one parent speaking the testing language at home, iv) provided detailed information about delivery, medical and

psychiatric history, and basic demographic information from both biological parents, v) lived with at least one biological parent. General exclusion criteria for the BATSS study were: i) diagnosis of epilepsy or history of fits/convulsions, ii) known presence of genetic syndrome related to autism, iii) known presence of significant uncorrected vision or hearing impairment, iv) premature birth, prior to week 34, v) presence of developmental or medical condition likely to affect brain development (e.g., Cerebral Palsy), vi) presence of twin-to-twin transfusion syndrome, vii) birth weight below 1.5 Kg.

When analysing missing data patterns, we did not consider infant excluded from the BATSS study overall, but rather infants who were in the BATSS study but for various reasons were excluded from any of the three analysed experiments. Twenty-eight infants were tested within the general BATSS study but were excluded prior to analysis of the current study as a detailed inspection of their records revealed that they did not fulfil the general criteria, as listed above. The final sample consisted of 594 infants (295 complete pairs, 164 monozygotic, and 4 incomplete pairs, 2 monozygotic; see Table 1 for demographics). This entails that more than 95% of the BATSS sample was included in the analysis.

### Experimental measures

We used data from three different experiments (gaze tracking, pupillometry, EEG; see Fig. 1) included in the BATSS study, collected via different machines but on the same testing day. For each experiment, we extracted 1) experiment-level missingness (whether the participant provided data for that experiment and hence was included in the original study), and 2) trial-level missingness (coded as data availability, i.e., how many trials were deemed usable according to the original studies, irrespective of whether the participant was included in the final analyses in the original research).

Data on experiment-level missingness were included for each participant in the final sample ($n = 594$) and coded for each experiment as a binary variable (i.e., 1 for missing data, 0 for valid data; see Fig. 1 c, g, k). Data was coded as missing (i.e., 1) when no data was provided (e.g., for lack of participation, technical reasons, or tiredness/fussiness), or when data was deemed of not good enough quality based on experiment-specific criteria (see definitions of 'poor data' linked to each experiment below). We chose to include in the analysis and code as missing the experimental sessions recorded by the research assistants as missing due to 'technical reasons' because: i) it can be difficult to know for sure whether a 'technical reason' is not at all linked to the child; ii) there were no major general technical issues with any of the experiments during data collection (e.g. periods of non-functionality); iii) while purely technical reasons will inflate E (if it affects one twin) or C (if it affects the pair), they cannot explain genetic influences detected in the data. Overall, missing data was observed for 21 infants reportedly due to technical reasons across any of the three experiments, counting for 3.5% of the total sample (i.e., 21/598 infants), and 8.8% (i.e., 21 infants with missing data for technical reasons/237 infants with missing data) of the experiment-level missingness.

In contrast to experiment-level missingness, trial-level missingness was a continuous variable (see details on each experiment below).

### EEG experiment (visual processing)

The infant/child EEG experiment has been used previously by us an others[18,23,24], and employs a combination of visual stimuli that form four experimental conditions: Global Form, Global Motion, Local Form, and Local Motion (activation pattern from the Global Motion condition is shown in Fig. 1b). Steady-state visual evoked potentials (SSVEPs) are examined in the visual cortex, as represented by the T2circ statistic[25], in response to the four stimulus conditions for a total experimental duration of approximately 3 minutes. EEG was measured using a 128-channel geodesic EEG net (EGI corp.) with standardized routines for infant measurements (sampling rate 500 Hz). See ref. 18 for a detailed description of the experiment.

In terms of experiment-level missingness due to poor data in the EEG experiment, 119 participants were excluded from the analysis entirely because visual inspection could not verify activation of the primary visual

**Fig. 1 | Experimental measures.** This figure illustrates the three experiments included in the current study: EEG (**a–d**); gaze tracking (**e–h**); pupillometry (**i–l**). Specifically, the experimental setup is shown in (**a, e, i**). The main experimental measure is shown in (**b, f, j**) (images were adapted from the original studies, respectively. see refs. 16–18). The distribution of data experiment-level data missingness is shown in (**c, g, k**). The distribution of trial-level data (indexing data quality) is shown in (**d, h, l**) as violin plot and boxplot (marking sample mean and quartiles), together with individual infant's data (black dots), and threshold for experiment-level inclusion (vertical dashed line in blue). The person shown in (**e, f**) provided consent for the use of the photograph.

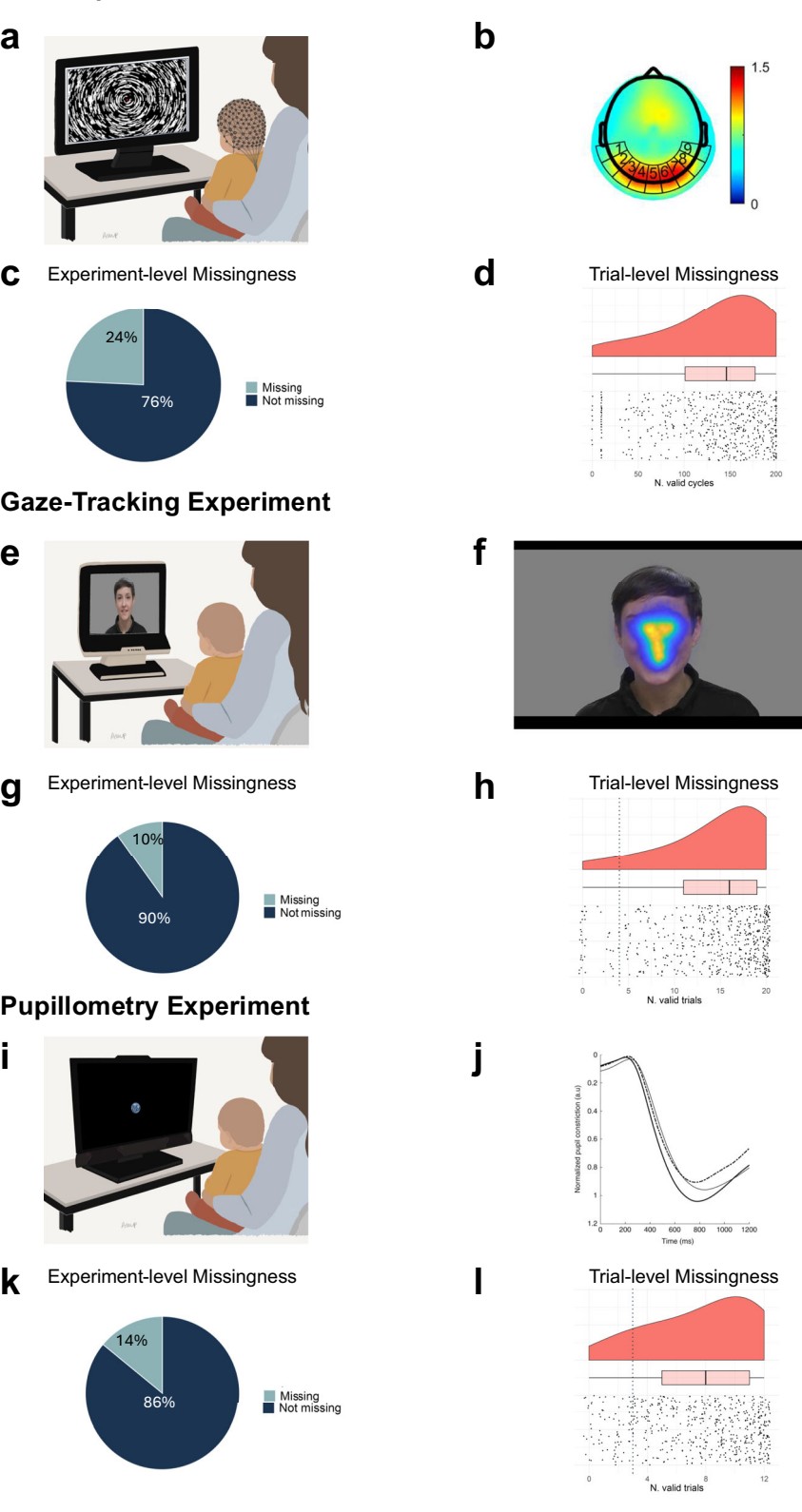

cortex to simple visual changes (data from Local Form condition, aggregated from whole session; stimulus duration 2 min), a response which was clearly visible in the majority of individual infants[18]. To test the reproducibility of this manual coding based on visual inspection of EEG data, we performed an inter-rater reliability test based on a subset of the participants' data (*n* = 100, randomly selected), by in independent rater blind to the purpose of the

study. Overall agreement was 92% (sensitivity (agreement good data) = 98%; specificity (agreement bad data) = 86%; Cohen's kappa = 0.84, 95%CI [0.73; 0.95]). In addition, 26 infants (18 MZ twins) in the sample selected did not provide any data due to: technical issues (1 DZ individual infant, 2 MZ individual infants, 2 MZ twin pairs, 2 DZ twin pairs); lack of participation (1 DZ individual infant, 1 MZ individual infant, 3 MZ twin pairs); fussiness (1

## Table 1 | Sample description

| Zygosity: | | DZ (n = 264) | MZ (n = 330) | p |
|---|---|---|---|---|
| **Age in days** | Mean (SD) | 167.6 (8.9) | 167.5 (8.6) | 0.831 |
| **Sex (n)** | Female | 132 | 152 | 0.383 |
| | Male | 132 | 178 | |
| **Gestation age in days** | Mean (SD) | 260.9 (7.6) | 257.9 (7.7) | < 0.001 |
| **Mean parental education (n)** | Primary school | 7 | 8 | 0.649 |
| | Secondary school | 24 | 40 | |
| | Tertiary - Undergraduate | 133 | 155 | |
| | Tertiary - Postgraduate | 100 | 127 | |
| **Family income (n)** | <20,000 SEK | 2 | 4 | 0.107 |
| | >100,000 SEK | 47 | 38 | |
| | 20–30,000 SEK | 8 | 10 | |
| | 30–40,000 SEK | 20 | 16 | |
| | 40–50,000 SEK | 26 | 46 | |
| | 50–60,000 SEK | 26 | 32 | |
| | 60–70,000 SEK | 38 | 49 | |
| | 70–80,000 SEK | 34 | 40 | |
| | 80–90,000 SEK | 28 | 61 | |
| | 90–100,000 SEK | 24 | 26 | |
| | unknown | 1 | 4 | |
| | (Missing) | 10 | 4 | |
| **Mean parental age in years** | Mean (SD) | 35.5 (4.9) | 35.1 (4.7) | 0.335 |
| Composite score (experiment-level) | Number of Missing | 91 | 142 | 0.041 |
| EEG | Number of Missing | 55 | 90 | 0.086 |
| Gaze tracking | Number of Missing | 27 | 32 | 0.939 |
| Pupillometry | Number of Missing | 35 | 49 | 0.664 |
| EEG: n valid trials | Mean (SD) | 134.9 (55.4) | 131.2 (55.5) | 0.425 |
| Gaze tracking: n valid trials | Mean (SD) | 14.9 (5.3) | 13.9 (5.7) | 0.028 |
| Pupillometry: n valid trials | Mean (SD) | 7.8 (3.6) | 7.6 (3.5) | 0.567 |

This Supplementary Table hows uncorrected *p*-value on differences across the sample stratified by zygosity: *MZ* monozygotic twin, *DZ* dizygotic twin.

DZ individual infant, 1 MZ twin pair, 1 DZ twin pair); tiredness (1 MZ individual infant); excessive movement (1 DZ individual infant). Trial-level data for these infants was coded as NA. Together with the ones excluded due to poor data, experiment-level missingness for the EEG experiment amounted to 24% of total sample; Fig. 1c).

Trial-level missingness was defined as the number of cycles (corresponding to pattern changes in the stimuli) that contributed with data for the analysis of visually evoked potentials in the Global motion condition (possible range: –240; Fig. 1d). Exclusion of data at this level exclusively reflected automatic rejection of cycles with a voltage range exceeding 100uV.

### Gaze tracking (preferential looking to eyes versus mouth)

This is a gaze tracking experiment designed to assess the infant's looking preference to the eyes versus the mouth region of a face[17]. Gaze data were recorded using a Tobii T120 Eye-tracker with a sampling rate of 60 Hz, using a standard Tobii monitor at native resolution (1,024 × 768) while the infants were presented with 20 stimuli videos in a pseudo-random order.

The videos comprised three conditions: Singing (12 videos of a woman singing common Swedish nursery rhymes); Talking (four videos of a woman saying common Swedish rhyme verses); and Still (four videos of a woman smiling). In all videos, a woman was centred in the video and the background was grey (there were two women, each of them contributing equally to all conditions). The length of the videos ranged from 4 to 12 s (for further details about the experiment, see ref. 17).

Missing data variables were extracted from this experiment based on the main experimental variable called 'the eye-mouth-index' (EMI), computed as the mean amount of gaze to the eye region, divided by the mean amount of gaze to both the eye and the mouth regions (see ref. 26). A total of 40 participants who provided less than 4 valid trials (stimulus videos) were excluded from the experiment (Experiment-level missingness due to poor data). In addition, 19 infants (10 MZ twins) in the sample selected did not provide any data due to: technical issues (1 MZ twin pair, 2 DZ twin pairs); lack of participation (1 MZ twin pair); fussiness (1 MZ individual infant, 1 DZ individual infant); tiredness (2 DZ individual infants); lack of time (1 MZ individual infant, 2 MZ twin pairs, 1 DZ twin pair). Trial-level data for these infants was coded as NA. Together with the ones excluded due to poor data, experiment-level missingness for the gaze-tracking experiment amounted to 10% of total sample; Fig. 1k.

Trial-level validity was defined based on a total looking time of at least 1 s to the screen combined with multiple cut-offs linked to the spatial distribution of gaze within the stimulus (Fig. 1l; see ref. 17 for details). The number of trials deemed valid for the EMI (trial-level missingness) could range between 0 and 20.

### Pupillometry (pupillary light reflex; PLR)

This is an infant-friendly eye-tracking experiment designed to elicit a pupillary light reflex (PLR, see refs. 16, 27), which is a physiological response to stimulus luminance serving to regulate the amount of light reaching the retina (Fig. 1f). Infants' PLRs to brief flashes of light were recorded using a TX-300 Tobii eye-tracker (120 Hz sampling rate, 23' screen unit with 1920 and 1080 resolution) from a maximum of 12 flash-stimuli (trials), embedded within a broader eye tracking battery with other types of stimuli (for details, see ref. 16).

Experiment-level missingness due to poor data for pupillometry was defined as providing less than 3 valid trials (61 infants). In addition, 23 infants (15 MZ twins) in the sample did not provide any data due to technical reasons (2 MZ twin pairs, 1 DZ twin pair), lack of time (1 individual MZ infant, 1 MZ twin pair), bad calibration (1 DZ individual infant), or tiredness (2 MZ twin pairs, 2 DZ twin pairs, 4 MZ individual infants, 1 DZ individual infant). Trial-level data for these infants was coded as NA. Together with the ones excluded due to poor data, experiment-level missingness for the pupillometry experiment amounted to 14% of total sample; Fig. 1g).

Trial-level missingness was defined as the number of valid trials (ranging between 0 and 24; Fig. 1h). Trials were deemed invalid based either on automated identification of artefacts and implausible responses in the time window for the PLR (Fig. 1f, see also ref. 27 for further details) or after visual inspection of the pupil time-series in each trial (researcher blind to zygosity). Trial data for each eye was considered separately (hence the maximum number of valid trials was 24, see ref. 16).

### Statistical analyses

The analysis plan including hypotheses and aims was pre-registered in the Open Science Framework (see ref. 22) after data collection and pre-processing, but prior to data analysis. Data from the three different experiments described above were processed and published in peer-reviewed journals before the pre-registration of this study.

### Twin analysis

Etiological structure was tested through univariate and multivariate twin modelling in three main analytic steps:

1. Univariate testing on experiment-level measures of data missingness across the three experiments, separately and as a composite (i.e., four different models tested, with sub-models, as specified below).
2. Univariate testing on trial-level measures of data missingness across the three experiments (i.e., three different models tested, with sub-models, as specified below).

For each measure, saturated models with nested models (testing the assumptions of equality of mean and variances across twin order and zygosity), as well as univariate twin models were fitted separately and reported in the Supplementary Material.

As noted above, as an overall measure of experiment-level missingness, we computed a binary composite measure indexing the presence of missing data from any of the three experiments. Based on this measure, a univariate liability threshold model[28] was used to decompose variability across individuals in experiment-level missingness into genetic and environmental components. In our pre-registered analysis plan, the overall analysis of experiment-level missingness was planned to be performed on an ordinal variable obtained from the sum of binary variables across experiments, leading to a three-level liability threshold model. Due to model convergence issues with the three-level liability threshold model, we decided to perform a single threshold model instead based on the binary variable obtained by collapsing ordinal levels above zero. Further, we performed a single threshold model to investigate inter-individual variability in experiment-level missingness separately for each experiment, yielding a total of four models tested for experiment-level missingness. See Supplementary Methods 1 for full equations.

Similarly, for trial-level missingness, we used classical univariate twin models to test validity of assumptions of equality of means and variances across zygosity and twin order for each of the three scores (Fig. 1d, h, l), transformed through a quadratic power to ensure skewness lower than 0.3, then scaled. In this case, three different ACE models (with sub-models) were tested, one for each experiment. See Supplementary Methods 1 for full equations and hereafter a more detailed explanation of the analytical steps performed to test sub-models.

An ACE model was used to decompose covariance among scales into additive genetics (A), shared environment (C), and unique environment factors (E), and tested against nested models (i.e., AE, CE, and E models). The best-fitting nested model was defined as the non-significant model with the lowest AIC value (Akaike Information Criteria; balancing variance explained and parsimoniousness of the model), to ensure that the model fit is not significantly poorer than the full model (as indexed by the $\chi^2$ distribution). This entails that the selected nested model is always statistically non-significant[29]. Once the nested model was selected, significance of the genetic (A) or shared environment component (C) was tested by comparing model fit through likelihood ratio test against the nested model with only E (e.g., AE model vs. E model), which corresponds to forcing the A or C term to zero. Components were considered significant if $p < 0.05/8 = 6.25 \ast 10^{-3}$ (based on 2 component estimates for each of the four models tested) based on Bonferroni correction for multiple comparison on the hypothesis of detecting significant familial influences on experiment-level data missingness. Similar corrections applied to trial-level data missingness, with the criterion translating to $p < 0.05/6 = 8.30 \ast 10^{-3}$ (based on 2 component estimates for each of the three models tested). This was not possible for the E term, for which we refer to the CI as an index of significance, in line with the standard reporting for twin research[30,31]. Whether or not the CI overlaps with zero shows whether the component is statistically significant. A similar approach was used to test significance of twin correlations by comparing the selected model with one in which the correlation of interest was set to zero. We used Bonferroni correction to test for multiple comparisons separately for hypotheses on experiment-level and trial-level data missingness, yielding respectively to following thresholds: $p < 0.05/8 = 6.25 \ast 10^{-3}$ (based on 2 correlations tested, one for monozygotic and one for dizygotic twins, across four models) and $p < 0.05/6 = 8.30 \ast 10^{-3}$ (based on 2 correlations tested across three models).

Data analysis was performed in R 4.1.2[32], and twin model fitting was performed through maximum likelihood optimization with the R package OpenMx, version 2.19.8[33].

## Associations between missingness and other traits
We also examined missing data patterns by testing associations between trial-level measures of missing data across the three experiments and measures obtained from questionnaires, representing the traditional and common way to evaluate systematic biases in missingness[5,7,34]. We tested associations with: age; sex; parental age; parental education; family income; gestational age; concurrent measures of adaptive functioning across social-communication and motor domains, as obtained from the Vineland Adaptive Behavior Scales (VABS[35] and detailed in ref. 36); concurrent measures of sensory-related behaviours, as measured across the four quadrants (i.e., sensory sensitivity, sensation avoiding, sensation seeking, and low registration) of the Infant/Toddler Sensory Profile (ITSP[37]); and concurrent measures of temperament, as measured by the three broad domains (i.e., negative affect, effortful control, and surgency) of the Infant Behavior Questionnaire (IBQ, short form[38]).

Furthermore, in line with our preregistration, we tested associations with autism, as indexed by genetic likelihood for developing the condition and level of autistic traits at 36 months of age. Genetic likelihood for developing autism was measured through polygenic scores, indexing the cumulative presence of common variants in the infant's DNA that are considered to be significantly associated to autism based on a reference genome-wide association study[39]. DNA samples were genotyped using Infinium Global Screening Array (Illumina, San Diego, CA, USA). Genotype quality control and processing followed standard procedures, as described in ref. 19. Genome-wide polygenic scores were calculated using polygenic prediction via Bayesian regression and continuous shrinkage priors method[40], using as reference statistics the most recent and largest genome-wide association studies for autism available at the time[39]. For the phenotypic association analysis with missing data scores, we included as covariates the first 10 principal components summarizing differences in genetic ancestry across individuals, as derived from genome-wide genetic data and commonly used to control for population stratification in genetic analyses.

Autistic traits at 36 months of age were measured by the Quantitative Checklist for Autism in Toddlers (Q-CHAT[41]), which is a 25-item parent-rated questionnaire assessing socio-communicative delays and sensor-imotor behaviours linked to autism development. Of note, the Q-CHAT total score showed 34% missing data, largely due to attrition. Missing data were imputed through multiple imputation based on a regression tree algorithm as implemented in the R package mice[42] and detailed in ref. 43.

We used the robust sandwich estimator in generalized estimating equations (GEE) to test phenotypic associations between trial-level missingness across the three experiments (dependent variables) and the demographic, questionnaire, and autism-related measures listed above (independent variables) while accounting for correlations between twins in a pair[44]. All dependent and independent variables were scaled so that the resulting Beta values were standardized. This procedure yielded 51 separate models tested in total, with Bonferroni correction used to correct for multiple comparisons adjusted for the independent variables (i.e., 0.05/17).

## Results
Sample demographics and descriptive statistics are shown in Table 1.

## Experiment-level missingness
A univariate liability threshold model[28] was used to decompose inter-individual variability in the presence of missing data in any of the experiments investigated into genetic and environmental components. The pre-specified[22] variable of interest was a binary composite measure indexing the presence of missing data (i.e., coded as 1 if the infant did not provide valid data for at least one of the three experiments [$n = 237$ infants, 39.6% of the total sample]; 0 if the infant provided valid data for all the three experiments

**Table 2 | Summary of findings from univariate twin models on experiment-level and trial-level missing data**

| Experiment-level missingness | | | | | | | |
|---|---|---|---|---|---|---|---|
| Model | AIC | Δ LL | Δ df | p | A | C | E |
| **Composite score** | | | | | | | |
| ACE | 787 | — | — | — | 0.19 [0; 0.60] | 0.22 [0; 0.52] | 0.59 [0.40; 0.79] |
| **CE** | **785** | **0.29** | **1** | **0.59** | — | **0.37 [0.20; 0.53]** | **0.63 [0.47; 0.80]** |
| **EEG** | | | | | | | |
| ACE | 654 | — | — | — | 0.42 [0; 0.64] | 0.02 [0; 0.52] | 0.56 [0.36; 0.80] |
| **AE** | **652** | **0.005** | **1** | **0.94** | **0.45 [0.22; 0.64]** | — | **0.55 [0.36; 0.78]** |
| **Gaze tracking** | | | | | | | |
| ACE | 381 | — | — | — | 0.03 [0; 0.75] | 0.46 [0; 0.70] | 0.51 [0.24; 0.79] |
| **CE** | **379** | **0.003** | **1** | **0.96** | — | **0.48 [0.21; 0.70]** | **0.52 [0.30; 0.79]** |
| **Pupillometry** | | | | | | | |
| ACE | 462 | — | — | — | 0.19 [0; 0.81] | 0.46 [0; 0.76] | 0.34 [0.17; 0.57] |
| **CE** | **461** | **0.27** | **1** | **0.60** | — | **0.62 [0.42; 0.77]** | **0.38 [0.23; 0.58]** |
| **Trial-level missingness** | | | | | | | |
| **EEG** | | | | | | | |
| ACE | 1597 | — | — | — | 0.34 [0.01; 0.46] | 0 [0; 0.25] | 0.66 [0.54; 0.80] |
| **AE** | **1595** | **−3.6*10⁻¹¹** | **1** | **1** | **0.34 [0.20; 0.46]** | — | **0.66 [0.54; 0.80]** |
| **Gaze tracking** | | | | | | | |
| ACE | 1624 | — | — | — | 0.27 [0; 0.39] | 0 [0; 0.22] | 0.73 [0.60; 0.87] |
| **AE** | **1622** | **−3.0*10⁻¹¹** | **1** | **1** | **0.27 [0.13; 0.39]** | — | **0.73 [0.61; 0.87]** |
| **Pupillometry** | | | | | | | |
| ACE | 1573 | — | — | — | 0.53 [0.32; 0.63] | 0 [0; 0.16] | 0.47 [0.37; 0.59] |
| **AE** | **1571** | **−1.4*10⁻¹⁰** | **1** | **1** | **0.53 [0.41; 0.63]** | — | **0.47 [0.37; 0.59]** |

The best-fitting univariate nested model is shown for each missing data score across both levels of analysis (experiment-level and trial-level) compared to the full ACE model. The best-fitting model was defined as the non-significant model (as indexed by the $\chi^2$ distribution) with the lowest AIC value (balancing variance explained and parsimoniousness of the model).
*AIC* Akaike Information Criterion, *ΔLL* difference in log-likelihood fit statistics from the reference model, *Δdf* difference in degrees of freedom from the reference model, A = % variance explained by additive genetics, C = % variance explained by shared family environment, E = % variance explained by unique environment.

[$n$ = 361 infants, 60.4% of the total sample]; see Fig. 1c, g, k). Twin correlations were significantly higher than zero for MZ twins (rMZ = 0.41; 95% confidence interval, CI = [0.18; 0.60]; $p$ = 5.2*10⁻⁴) while they did not survive Bonferroni corrections for DZ twins (rDZ = 0.32; CI = [0.04; 0.56]; $p$ = 0.03).

Twin modelling assumptions of equality of mean and variances across twin order and zygosity were met (see Supplementary Table 1 for details), and univariate ACE models showed a contribution of shared and unique environment to inter-individual variability in the overall liability for missing data (CE best-fitting models; see Supplementary Table 2 for details). In particular, the composite score showed moderate influences from shared environment (C = 0.37, CI: [0.20; 0.53]; $p$ = 4.3*10⁻⁵), and substantial influences from unique environment (E = 0.63, CI: [0.47; 0.80]), while genetic influences were not significant ($p$ = 0.59).

For completeness, we also performed equivalent analyses on the three experiments individually (analysis not pre-specified). All twin modelling assumptions were met (see Supplementary Tables 3–5 for details), and univariate ACE models showed a contribution of shared and unique environment to inter-individual variability in the liability for missing data in gaze tracking (C = 0.48, CI = [0.21; 0.70], $p$ = 6.2*10⁻⁴; E = 0.52, CI = [0.30; 0.79]) and pupillometry experiments (C = 0.62, CI = [0.42; 0.77], $p$ = 5.7*10⁻⁸; E = 0.38, CI = [0.23; 0.58]; CE best-fitting models; see Supplementary Table 7-S8 for details), while additive genetics and unique environment explained variability across individuals in the liability for missing data in the EEG experiment (A = 0.45, CI = [0.22; 0.64], $p$ = 2.0*10⁻⁴; E = 0.55, CI = [0.36; 0.78]; AE best-fitting models; see Supplementary Table 6 for details).

### Trial-level missingness

Trial-level missingness was indexed by the number of experimental trials available in the PLR and gaze tracking experiments, and cycles available for analysis in the EEG experiment (i.e., trial-level data availability, see Methods). Prior to formal twin modelling, we checked that patterns of twin similarity were reasonable and supported the general twin modelling approach. Correlations for MZ twins were significant but moderate-to-low for the gaze tracking (i.e., rMZ = 0.29, CI = [0.15; 0.42]; $p$ = 7.0*10⁻⁵) and the EEG experiment (i.e., rMZ = 0.35; CI = [0.20; 0.47]; $p$ = 5.2*10⁻⁶), while the pupillary experiment showed moderately high correlations (i.e., rMZ = 0.55; CI = [0.44; 0.65], $p$ = 4.8*10⁻¹³). Correlations for DZ twins were not significantly different from null for the gaze tracking (rDZ = 0.05; CI = [−0.14; 0.22]; $p$ = 0.69) and EEG experiments (rDZ = 0.12; CI = [−0.05; 0.29]; $p$ = 0.19), and did not survive Bonferroni correction for the pupillometry experiment (i.e., rDZ = 0.17; CI = [0.008; 0.32]; $p$ = 0.04). Twin modelling assumptions of equality of mean and variances across twin order and zygosity were met (see Supplementary Tables 9–11 for details).

Variability in trial-level data missingness across the three experiments was explained by additive genetics and unique environment explained (i.e., AE model solution; see Table 2), as indicated by the estimates of variance decomposition across the different scores (equivalent to univariate estimates). Specifically, we observed moderate influence from additive genetics on variance in trial-level missingness obtained from the gaze tracking experiment (A = 0.27, CI = [0.01; 0.39]; $p$ = 1.2*10⁻⁴) and the EEG experiment (A = 0.34, CI = [0.02; 0.46]; $p$ = 2.6*10⁻⁶), and large genetic influences for trial-level missingness in the pupillometry experiment (A = 0.53; CI = [0.33; 0.63]; $p$ = 1.4*10⁻¹³). Accordingly, unique environment had a large effect on trial-level missingness in the gaze tracking and EEG experiments

(E = 0.73, CI = [0.61; 0.87]; E = 0.66, CI = [0.54; 0.80], respectively) and a moderate effect in the pupillometry experiment (E = 0.47; CI = [0.37; 0.59]).

We found little covariance across the quantitative measures indexing missing data in the three experiments. Specifically, the phenotypic correlation was significant but low between the gaze tracking and the pupillometry experiment (r = 0.19, CI = [0.09; 0.27], $p = 2.6*10^{-5}$), while it was not significant between gaze tracking and EEG experiments (r = −0.01, CI = [−0.10; 0.07], p = .80), or between pupillometry and EEG experiments (r = 0.006, CI= [−0.08; 0.09], p = 0.89). Cross-twin cross-trait correlations followed an analogous pattern, with low MZ correlation between quantitative scores from the pupillometry and gaze tracking experiments (r = 0.12, CI = [0.01; 0.22]; p = .03) which did not survive Bonferroni corrections for multiple comparisons (with threshold $p < 5.6*10^{-3}$ based on 9 comparisons tested), and DZ correlations also not significant (r = 0.04, CI = [−0.09; 0.16]; p = 0.53). The cross-twin cross-trait correlation between scores from the EEG experiment and any of the two eye-tracking experiments were not significant for any zygosity groups.

### Associations between missingness and other traits

Following the traditional approach for evaluating the extent to which missingness is systematic, we tested associations between trial-level measures indexing data quality across the three different experiments and demographic measures; concurrent questionnaire data measuring adaptive functioning, temperament traits, and sensory-related behaviours; and measures of genetic likelihood for autism and autistic traits in toddlerhood. Descriptives are shown in Supplementary Table 12.

After applying stringent criteria for multiple testing, we found no significant association between trial-level measures of data quality and demographic, genetics, or behavioural data (see Supplementary Table 13 for details).

### Discussion

Understanding the nature of data missingness is essential for developing effective strategies to address it. This includes gauging the magnitude of systematic influences to missingness and their variation across different data types. We examined such influences across three experiments from developmental cognitive neuroscience by investigating etiological influences, such as genetic factors and shared or unique environmental factors, on variability across individuals in missing data. For both experiment-level and trial-level missingness, we adhered to the criteria originally used in the respective studies[16–18].

Our findings show significant familial influences (from shared environment and additive genetics) on missing data in all three experiments across both levels of analysis. For some types of missingness, more than half of the variance was attributable to genetic factors (Pupillometry, trial-level missingness). The original work on the experiments investigated here[16–18] showed that many of the variables involved in these experimental tasks are themselves heritable, which raises the possibility that shared genetic factors may influence both performance and the likelihood of missingness.

Sixty percent (60.4%) of infants were included in the data analysis in all experiments, while approximately forty percent (39.6%) were missing from at least one experiment. Variance in this dichotomous measure of overall experiment-level missingness was influenced by shared environment (speculatively, this may include common variation in experimental procedures that day, same researcher collecting the data, but also distant and 'classical' environmental factors linked to e.g., the home environment) and unique environment (speculatively, this could include individual-specific technical errors, or unique experiences from the environment on the testing day or before). It is notable that the unique environment component includes 'measurement error' which takes a more nuanced meaning when the variable of interest is data missingness. Simplified, measurement error can be thought of as non-reproduceable intra-individual variation in data missingness; however, even completely idiosyncratic and non-reproduceable events may be meaningful and fully interpretable in this context (e.g., someone entering the room by mistake, disturbing the recording).

Follow up analyses focusing on each experiment revealed that being excluded from the EEG experiment was a moderately heritable phenotype (Fig. 1c), while being excluded from eye tracking-based experiments (PLR and gaze tracking) was linked to shared environment (Fig. 1g,k). Previous work has looked into specific sources of data loss in eye-tracking studies[8,21,45,46], which can be categorized into technical problems in the equipment (e.g., lack of precision or accuracy, missed recording), choices made by the testing researcher (e.g., positioning), and participant characteristics (e.g., eye colour, age, or being more fidgety). Data loss in experiments with 5-month-old infant participants may be particularly influenced by the researcher who conducted the testing[8], representing likely contributors to the environmental influences observed in our study. Similar influences of the experimenter on data loss have been shown before for infant research using EEG[9]; however, we found no significant influences of shared environment on data loss for the EEG experiment.

Our analyses on trial-level data missingness, indexing the amount of available data and hence an important aspect of data quality, pointed to influences from additive genetics across all three experiments. The influence of shared environment was non-significant. This shows that individual differences rooted in the child's biology impact data quality across several different experimental assessments of brain and behaviour. This finding may explain previous findings of within-participant stability in the percentage of data loss across multiple longitudinal sessions in infant experiments, particularly in relation to EEG[9].

Whether a child provided good-quality data (low trial level missingness) in one experiment was found to be largely unrelated to that child's data quality in the other experiments. It is likely that the experiment-specific rules and cut-offs for trial exclusion contributed to the generally low covariance between the observed measures. Yet, this finding is surprising considering the many commonalities between the experiments examined here and did not support our pre-registered hypothesis that there would be positive associations in trial-level missingness across different experiments. Specifically, all three experiments taxed infants' attention, and generally poor attention spans and "fussiness" would be expected to impact not only one experiment. Further, the experiments were conducted in the same day and, for most of infants, by the same experimenter. The low but significant covariance detected was indeed between gaze tracking and pupillometry experiments, likely due to the overlapping methodology (eye tracking in both, yet different eye tracker models). Arguably, of the six missingness indexes included in this study, the trial-level missingness for EEG was most distinct, as it reflected automatic rejection by the EEG software if voltage range exceeded 100 uV.

Previous studies have indicated a link between missing data and neurodevelopmental conditions such as autism[11], but we did not find significant association with genetic background for autism or autistic traits at 36 months of age (Supplementary Table 13). This suggests that the amount of data provided in experimental settings in early infancy might not reflect autism-related genetics or autistic traits later in toddlerhood. However, we note that the sample examined here is a general population sample, and such associations may exist among infants at elevated familial likelihood of developing autism. Applying stringent criteria, no significant associations were detected between data quality and demographic or child behaviours (Supplementary Table 13). However, we note that statistical power to detect small effects, particularly after stringent correction for multiple testing, may have been limited. Therefore, the absence of significant associations should be interpreted cautiously, and future studies with larger samples or enriched designs (e.g., elevated-likelihood cohorts) are needed to further assess these relationships.

Our results indicate that genetic factors contribute to variation in data missingness, which may imply that the common (and often implicit) assumption of missing data being completely at random should not be made uncritically in similar experiments within developmental cognitive neuroscience. The observed inter-individual differences in data missingness, partly influenced by genetic factors, raise the possibility of statistical dependence between missingness and traits of interest. As such, relying on

procedures that assume MCAR (e.g., listwise deletion) may introduce bias and limit the generalizability of findings. More advanced methods to deal with missing data provide different solutions by leveraging on the association with observed variables to estimate missing values (e.g., multiple imputation approaches[6,43]) and/or distribution parameters (e.g., maximum likelihood strategies[47]). While the use of such methods has not been widely adopted by developmental scientists, it would allow researchers to exploit the power of the collected dataset more fully, providing more confidence in the observed patterns and likely increasing reproducibility. This has a particular impact on those analyses integrating different data modalities, like eye-tracking and neural, electrophysiological measures with clinical and questionnaire data, in which complete-case analysis has even a stronger influence. Finally, showing the influence of familial, including genetic, factors on inter-individual differences in data missingness provided across different experimental assessments of brain and behaviour, this study urges to move away from the general consideration of missing data as noise and to explore new strategies to detect potentially meaningful signal in the data.

## Limitations

The results of this study reflect the specific studies selected and criteria they used to define validity of data at a trial and experiment level. While the studies reflect common methods in developmental cognitive neuroscience (i.e., gaze tracking, pupillometry, and EEG), and have all been published in peer-reviewed journals[16–18], future work is needed to evaluate generalizability of findings to different experimental settings and at other ages. Notably, while the experiment-level missingness for EEG was based on visual inspection, these decisions had a substantial to near excellent inter-rater reliability (see Results). This reflects the fact that these phasic stimuli, if one attends to them, create a clear response in the visual cortex at the primary stimulated frequency. Next, while the results of this study are informative of factors influencing data missingness among the infants who partook in the study, they do not tell us about factors influencing research participation more broadly (choosing not to take part or being excluded due to general exclusion criteria). Finally, as noted above (see Methods), there were deviations from the preregistration.

## Data availability

The study used behavioural data and derivates of genetic testing. The data contain pseudonymized personal information as defined by GDPR (EU law) and cannot be openly shared. Access is available through a controlled access procedure for researchers who: (a) demonstrate a legitimate scientific purpose aligned with the original participant consent and ethics approval; (b) obtain approval from their institutional ethics board; (c) have the capacity to ensure GDPR-compliant secure data handling; and (d) sign a Data Sharing Agreement specifying conditions for use, storage, and destruction of the data. To request access, contact the corresponding author (TFY). The numerical data underlying Fig. 1 are available here: https://osf.io/bg8zx/files/osfstorage.

## Code availability

The analytic code necessary to attempt to replicate the findings presented here is publicly available at https://doi.org/10.5281/zenodo.19333394.

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

## Acknowledgements

The authors thank all participating families, as well as researcher Dr Pär Nyström and research assistants Linnea Hamrefors, Joy Hättestrand, Lynnea Myers, Johanna Kronqvist, Sofia Jönsson, Anna Kernell, Carolin Schreiner, Sophie Lingö, Angelinn Liljebäck, Isabelle Enedahl, Matthis Andreasson, Lisa Belfrage, Mattias Savallampi, Isabelle Ocklind and Hjalmar Nobel Norrman. The genotyping was done at the SNP&SEQ Technology Platform, Uppsala University, and PGS calculation supported by Dr. Danyang Li and Dr. Kristina Tammimies. This research was funded by Riksbankens Jubileumsfond, the Knut and Alice Wallenberg Foundation, the Marianne and Marcus Wallenberg Foundation, and the Innovative Medicines Initiative 2 Joint Undertaking (grant number 777394; supported from the European Union's Horizon 2020 research and innovation programme and EFPIA and AUTISM SPEAKS, Autistica, SFARI). Any views expressed are those of the author(s) and not necessarily those of the funders. The funders had no role in study design, data collection and analysis, decision to publish or preparation of the manuscript.

## Author contributions

This study was conceptualized by G.B and T.F.Y. Data from the different experiments were pre-processed by A.M.P., C.V., and I.H., and analysed by G.B. G.B. drafted the manuscript, with contributions from T. F. Y., which was further refined and approved by all authors before submission.

## Funding

## Competing interest

The authors declare no competing interests.
