## [Transparent Peer Review file · Communications Psychology]

Genetic and environmental influences on data missingness in developmental cognitive neuroscience

Corresponding Author: Professor Terje Falck-Ytter

Version 0:

Decision Letter:

Dear Professor Falck-Ytter,

Thank you for your patience during the peer-review process. Your manuscript titled "Familial influences on data missingness in developmental cognitive neuroscience" has now been seen by 3 reviewers, whose comments are appended below. You will see that they find your work of some potential interest. However, they have raised quite substantial concerns that must be addressed. In light of these comments, we cannot accept the manuscript for publication, but would be interested in considering a revised version that fully addresses these serious concerns.

We hope you will find the Reviewers' comments useful as you decide how to proceed. Should additional work allow you to address these criticisms, we would be happy to look at a substantially revised manuscript. If you choose to take up this option, please highlight all changes in the manuscript text file, and provide a detailed point-by-point reply to the reviewers.

The reviewers raised several important concerns. First, we ask that you respond to the concerns by providing a clearer rationale for the selection of the three experimental paradigms (EEG, pupillometry, gaze tracking), including more detail on their representativeness and modality-specific sources of missingness, as well as the sample's relationship to the BATSS study. The introduction should provide a clear justification for the study's hypotheses and explain the proposal why genetic influences violate the MCAR assumption. Please address Reviewer 2's concerns regarding the conceptual interpretation of MCAR, through appropriate additional analyses and a sharpening of the conclusions.

We also ask that you improve the transparency in statistical modeling, and include more information in the main manuscripts, rather than in the SI. With regard to the preregistration, please ensure that your revision is fully in line with the journal's preregistration policy (<https://www.nature.com/commpsychol/editorial-policies/preregistration-policy>). All preregistered analyses should be reported, unless scientifically unfeasible or unsound. Deviations need to be clearly flagged and justified. Please address all methodological concerns, including those pertaining to visual trial exclusions in EEG, handling of composite scores and covariates (e.g., parental education), low power, and multiple testing.

I am attaching a checklist that details critical reporting requirements for the revised manuscript. Please attend to each item and ensure your manuscript is fully compliant. We are requesting that your manuscript aligns with these requirements as this facilitates the evaluation of your manuscript, reducing delays in re-review and potential future acceptance. If your revised manuscript is not aligned with these requests on major issues, such as those concerning statistics, it may be returned to you for further revisions without re-review. Additional information can be found in our style and formatting guide Communications Psychology formatting guide.

If the revision process takes significantly longer than five months, we will be happy to reconsider your paper at a later date, provided it still presents a significant contribution to the literature at that stage.

Please use the following link to submit your

- revised manuscript,
- point-by-point response to the referees' comments,
- cover letter (as a separate document),
- the Editorial Policy Checklist (see below),
- the Reporting Summary (see below), and
- the completed Editorial Request Table (attached):

Link Redacted

Thank you for the opportunity to review your work.

Best regards,

Anna-Lena Schubert

Anna-Lena Schubert, PhD
Editorial Board Member
Communications Psychology
orcid.org/0000-0001-7248-0662

REVIEWER EXPERTISE:

Reviewer #1: developmental psychology, EEG

Reviewer #2: missing data

Reviewer #3: twin-study design

REVIEWER REPORTS:

Reviewer #1 (Remarks to the Author):

This study examined systematic missingness of data across three infant experiments using a twin-design. A substantial amount of additive genetics and shared environmental effects was found, which was uncorrelated with additional data collected for characterization of participants (e.g., sex, age, or temperament).

The manuscript is overall very well written, provides important data and a valuable reference to the literature. However, some aspects require clarification in my view.

The intro is extremely short, thereby lacking any specifics regarding missing data in infant studies. While I recognize the need for writing concisely, the problem of missingness in infant studies needs to be characterized based on empirical reports. This also goes for the three experiments selected for this analysis. It should be stated clearly why these were chosen and what, if any, differences between data missingness can be expected. It seems likely that different genetically determined traits contribute to data missingness depending on the method, that is, hair quality and scalp shape might induce bad data quality in EEG, whereas eye physiology might contribute to data loss in eyetracking (EEG: Etienne et al., 2020, Annu. Inf. Conf. IEEE Eng. Med. Biol. Soc., Choi et al., 2021, Affect. Sci; eyetracking: Hessels et al., 2014, Infancy).

Moreover, the preparation demands are completely different between EEG and eyetracking, another potential source for differential missingness.

Also, in the Results section, the selected experiments are referred to as „representative“. While this term will be debatable in any case, I would agree that eyetracking of looking patterns for human faces is a standard paradigm. However, the EEG and pupillometry experiments run here do not seem representative, but rather highly specific instantiations of these methods. Of course all experiments required a minimum of attention, but attention was not a common DV among experiments. In particular, the experiment referred to as „EEG“ seems to have been an SSVEP experiment, which is not stated clearly. For any novices, this is important information which may also relate to data missingness and thus should be made explicit. The manuscript is not dealing enough with the specifics of the examined experiments and DVs in my view, thus leaving much room for speculation regarding the specificity of familial effects per experiment.

The preregistration mentions an exploratory research question, which should also be introduced at the end of the intro („We will also evaluate different multivariate models linking the different missingness indexes, but have no a-priori assumption about what kind of structure will be best fitting due to lack of previous empirical studies.“).

While I understand that data preprocessing here is based on the already published primary reports, the selection of trials based on visual inspection in FPVS is not objective and not state of the art. Responses at the stimulated frequency can be quantified as SNR or Z-scores, allowing for an objectified exclusion criterion, which in turn might be related differently to data missingness.

The preregistration described that the medical and psychiatric history information from both parents would be collected. Why was this information not used in the analyses? It might be this background information is related to data missingness, as it might (for example) reflect genetic variance.

Overall, please state very clearly how you deviated from the preregistration and why this was the case.

The discussion section mentions low power for detecting covariate effects. This should be substantiated by a power analysis.

Please specify how parental education was coded for analysis.

Reviewer #2 (Remarks to the Author):

This paper reports results from an analysis of genetic and environmental contributions to data missingness in a Swedish study of infant (5-6 months) twins in widely used 3 experimental paradigms to assess infant brain and behavioral development (EEG, pupillometry and gaze tracking). As I understood, the basic research aim was to evidence that the missing-completely-at-random (MCAR) assumption is typically not valid in this kind of research, as in particular genetic impacts cause missingness, which thus does not occur „randomly“ and statistical analyses relying on MCAR missing-data treatment may be biased. I have to admit that I am not an expert in infant research, hence cannot say whether it indeed suffers from unreflected and unjustifiable use of MCAR methodology, but if so, it would be an important methodological topic for this field of research. However, I have a serious concern regarding the authors' conceptual understanding of MCAR (and the respective theory of “missingness mechanisms” grounded on Rubin’s [1979] seminal work):

(1) Major point: IMO, this study’s general “rationale” is affected by a quite common misunderstanding of MCAR (as well as MAR and MNAR), considering these concepts as kind of causal theory of data missingness (see, e.g., Schafer & Graham, 2002, p. 151: “... Rubin’s (1976) definitions describe statistical relationships between the data and the missingness, not causal relationships”). That is, MCAR etc. are stochastic concepts, regarding the associations between the scores/values and the missingness of the variables analyzed with a statistical model. Thus, neither does knowledge of a cause of missingness in the assessments of a variable strictly imply that it cannot be MCAR whenever the variable is used in any statistical analysis, nor is knowledge of such causes needed to prove the MCAR does not hold. Given a set of variables (i.e., a multivariate distribution to be analyzed with a statistical model), MCAR means statistical independence of a variable’s missingness from all these variables (i.e., the observed and unobserved scores). Therefore, having evidenced that interindividual variation of missingness is partly due to genetic variation, the conclusion that “we can generally reject the common (explicit or implicit) assumption of data being missing completely at random” (page 8, line 177-179) is not warranted offhandedly. What would be needed in addition is some evidence that indeed the same (or similar) genetic impacts that increase the missingness probability do also affect the scores in the respective variables or at least some other variables included in a statistical analysis model, hence causing statistical associations between the missingness and the data. The Rubin concepts of missingness mechanisms imply, that these are specific to the model/set of variables analyzed (e.g., adding variables predictive of another variable’s missingness may shift the latter from MNAR to MAR). Thus, I strongly recommend (1) to “downtone” and prevent the notion of causality when referring to concepts such as MCAR (which seems particularly important as this causal misconception indeed is widespread among applied researchers), and (2) to strengthen and clarify the relevant insights to be gained from the analyses/results reported – my above critique did not mean to say that the disclosure of genetic variance “causing” missingness is not a relevant scientific goal here, I just doubt that this disclosure is generally relevant with respect to the application of MCAR vs. MAR or MNAR methods for the analysis of these measures. Moreover, hasn’t it by now long been accepted that at least MAR missing data treatment is always preferable to MCAR in presence of substantial shares of missing data – would researchers really need evidence of genetic impacts on missingness (even if this would be relevant with regard to MCAR) to do so?

(2) Given the rather general specification of a hypothetical prediction to be tested in this study (p. 3-4, l. 83-87), I wondered about the multiple testing involved in the analyses (nested model comparisons, tests of trial level missingness): What is/are the familywise error rate/s here, potentially needing some adjustments for multiple testing? If the study hypothesis indeed is just that some genetic effect will be found on some of the statistical tests, multiple testing adjustments would of course be needed. Thus, the study hypotheses should be stated more exactly and specifically. Also, as explained in my above point (1), I doubt that the prediction made on the bottom of p.3 is “contrary to the MCAR hypothesis”.

(3) I found it difficult to follow your analytical strategy without a more detailed and clear-cut presentation of the statistical models. Of course, readers of this manuscript may/should already know what a liability threshold model is and/or may gain such knowledge from reading respective methodological literature, but I doubt that this method is as well known as the most widely used regression models. It would facilitate reading if one could just somehow see the exact model used here – e.g. the model equations including the variables/measures modelled. This may, however, already have been presented in the Supplementary Material – I couldn’t open some of these pdf-files.

(4) Minor point, related to (3): “All twin modelling assumptions were met” (p. 5, l.101) – appeared to me a bit confusing, as one may think of modelling assumptions such as multivariate normality of the joint twin pair liability distribution (e.g. Benchek & Morris, 2013, Hum. Genet., doi: 10.1007/s00439-013-1334-z – if this is the analytic model you used). BTW, this latter study suggests that the ACE liability threshold model is highly vulnerable to violations of the MVN assumption – maybe also worth some consideration for your study?

(5) Minor point: On bottom of p. 2 (l. 57) and also later in the manuscript when you mention “individual variability”, I wondered whether you mean inter- or intra-individual variability (or both). Maybe some clarification here?

(6) P. 21: “We chose to include reasons of missingness recorded by the research assistants as ‘technical reasons’ because ... in the data” – I couldn’t understand what this means to say. E.g.: You call something technical reason “because it is difficult to know for sure whether a ‘technical reason’ is not at all linked to the child”? Could this be explained more clear-cut? I may, however, just have been slow on the uptake here.

Reviewer #3 (Remarks to the Author):

The paper investigated the genetic and environmental etiology of data missingness in infant twins and found that, while experiment-level missingness was influenced by shared and non-shared environmental factors (with the exception of EEG), trial-level missingness was influenced by genetic and non-shared environmental factors. Given that the etiology of data missingness has been rarely studied, the results provide a valuable contribution to the literature on this topic. Below are my comments and suggestions for improvement:

1. Introduction: Apparently, The BATSS dataset includes a wide range of data from infant twins. However, it is unclear why the authors specifically chose three variables—EEG, pupillometry, and gaze tracking—to study the etiology of data missingness. It is possible that other infant data may exhibit different etiological patterns. While the reason for this selection is briefly mentioned in the Discussion section, I suggest providing a more detailed and clear explanation for why these three variables were chosen in the Introduction.

2. Introduction: In the last paragraph, the authors hypothesized that genetic and shared environmental influences would affect the missing data, contrary to the assumption of missing completely at random (MCAR). However, the rationale for this hypothesis is not clearly explained in the Introduction. Providing a justification for this hypothesis in the Introduction would strengthen the paper and give readers a better understanding of the basis for this assumption.

3. Title: The term "Developmental Psychology" is included in the title of the paper. However, Developmental Psychology is a broad discipline that encompasses various subfields, including epidemiological research and studies on older adults. Given the focus of this study, I am concerned that the results may not be easily generalized to the field of Developmental Psychology as a whole. It may be more appropriate to specify a more focused research area, such as neurodevelopmental infant research, in the title.

4. Line 72- unique environmental influences include “measurement error”. Please specify this and explain what “measurement error” means in this particular study.

5. Sample- While the reference to the sample and the inclusion and exclusion criteria have been provided, there seems to be a lack of detailed information regarding the recruitment process. It would be helpful for readers to understand how the authors recruited participants for the study. Specifically, were participants recruited from a larger database, or was the sample selected from a particular subset of the population? Providing details on the recruitment process would offer transparency and allow readers to better assess the generalizability of the findings.

6. Additionally, it might be useful to state the general aim of the BATSS (presumably the larger study or dataset from which the sample was drawn) to give readers context for the sample’s characteristics.

7. It would be important to clarify whether the current sample includes all twins within the BATSS dataset or only those who participated in the specific experiments referenced. Understanding whether the sample is representative of the entire BATSS cohort or restricted to those who volunteered for certain experiments would help clarify the scope of the findings and their applicability.

8. Table 1: The number of missing data is consistently higher, and the number of valid trials is consistently lower in MZ twins compared to DZ twins, although these differences did not always reach statistical significance. Nevertheless, the consistent patterns are intriguing. Do the authors have any speculation for these patterns?

9. Please state how you calculated the composite score (Table 2).

10. The three variables are not highly correlated, and the sample size is relatively small. As a result, the study is underpowered to conduct a multivariate analysis. The multivariate analysis section introduces some confusion and unnecessary complexity. I suggest that the authors remove the multivariate twin analysis from both the manuscript and the supplementary information, and instead report the correlations among the three variables with discussion in the Results or Discussion section to avoid confusion.

EDITORIAL POLICIES

We ask that you ensure your manuscript complies with our editorial policies and reporting requirements.

To that end, we require revised manuscripts to be accompanied by two completed items: a reporting summary that collects information on study design and procedure, and an editorial policy checklist that verifies compliance with all required editorial policies

- <https://www.nature.com/documents/nr-reporting-summary.zip>>Nature Research Reporting Summary
- <https://www.nature.com/documents/nr-editorial-policy-checklist.pdf>>Editorial Policy Checklist

All points on the policy checklist must be addressed. Your revised manuscript can only be sent back to the referees if these checklists are completed and uploaded with the revision.

Notes: If you have submitted a Stage 1 Registered Report, Review, Primer, Comment, or Perspective you do not need to submit these forms. If you have already submitted these forms, you may disregard this request.

** Visit Nature Research's author and referees' website at <http://www.nature.com/authors>>www.nature.com/authors for information about policies, services and author benefits**

If you experience problems in linking your ORCID, please contact the <http://platformsupport.nature.com/>>Platform Support Helpdesk.

Version 1:

Decision Letter:

Dear Professor Falck-Ytter,

Thank you for submitting a revised version of your manuscript titled "Familial influences on data missingness in developmental cognitive neuroscience". After careful consideration and discussion with my colleagues, I am sorry to have to tell you that we do not feel that manuscript has been sufficiently revised to justify sending this revision back to the reviewers.

We take this unusual course of action is taken occasionally in order to avoid unproductive rounds of review that ultimately reduce the chances of the manuscript obtaining a fair and objective evaluation.

In order for us to consider this manuscript further please do your best to fully address all of the following issues: the EEG (FPVS) trial exclusion criteria.

Reviewer 2 raised a specific concern that visual trial exclusion is not objective or state-of-the-art, and that responses at the stimulation frequency can be quantified (e.g., SNR, Z-scores), enabling an objective exclusion criterion. Your current response ("we acknowledge this in the Discussion as a limitation") is not sufficient. For the present study, the absence of objective, signal-quality-based exclusion rules introduces experimenter degrees of freedom that can systematically bias

downstream estimates (including heritability) and can also change the nature of “missingness” itself.

To address this, please do one of the following:

(1) Re-analyze using objective thresholds for EEG trial/participant validity (e.g., SNR- or Z-based criteria at the stimulated frequency), and report how conclusions (including missingness patterns and heritability estimates) change under these principled rules.

or, if you maintain visual exclusion:

(2) Report inter-rater reliability for visual exclusion decisions (trial-level and/or participant-level as applicable) at least for a subset of randomly sampled epochs, including who rated, blinding procedures (if any), and reliability statistics (e.g., ICC/kappa, plus decision rules for disagreements). In addition, we would ask you to further elaborate on the limitations to generalization due to the reliance of visual exclusion and how it can affect the nature of missingness itself.

If you are able to adequately respond to these concerns, we would be happy to look at a revised manuscript; otherwise your best course of action may be to submit elsewhere.

We hope to receive your revised version as soon as possible. If you anticipate a delay of more than three months, however, please let us know. We will be happy to consider your revision so long as nothing similar has been accepted for publication at Communications Psychology or published elsewhere.

Please use the link below to submit a suitably revised manuscript and response to referees when they are ready.

Link Redacted

** This url links to your confidential home page and associated information about manuscripts you may have submitted or are reviewing for us. If you wish to forward this email to co-authors, please delete the link to your homepage first **

If you are not interested in submitting a suitably revised manuscript in the future please let me know immediately so we can close your file. If you have any questions, please contact me.

I am sorry that we cannot be more positive as things stand.

Best regards,

Anna-Lena Schubert

Anna-Lena Schubert, PhD
Editorial Board Member
Communications Psychology
orcid.org/0000-0001-7248-0662

Communications Psychology is committed to improving transparency in authorship. As part of our efforts in this direction, we are now requesting that all authors identified as ‘corresponding author’ create and link their Open Researcher and Contributor Identifier (ORCID) with their account on the Manuscript Tracking System prior to acceptance. ORCID helps the scientific community achieve unambiguous attribution of all scholarly contributions. You can create and link your ORCID from the home page of the Manuscript Tracking System by clicking on ‘Modify my Springer Nature account’ and following the instructions in the link below. Please also inform all co-authors that they can add their ORCID to their accounts and that they must do so prior to acceptance.

Version 2:

Decision Letter:

Dear Professor Falck-Ytter,

Your manuscript titled "Familial influences on data missingness in developmental cognitive neuroscience" has now been seen by our reviewers, whose comments appear below. In light of their advice I am delighted to say that we are happy, in principle, to publish a suitably revised version in Communications Psychology.

We therefore invite you to revise your paper one last time to address the remaining concerns of our reviewers and a list of editorial requests. At the same time we ask that you edit your manuscript to comply with our format requirements and to maximise the accessibility and therefore the impact of your work.

EDITORIAL REQUESTS:

SUBMISSION INFORMATION:

OPEN ACCESS:

* DATA AVAILABILITY:

Link Redacted

Best regards,

Marike Schiffer, on behalf of

Anna-Lena Schubert

Anna-Lena Schubert, PhD
Editorial Board Member
Communications Psychology
orcid.org/0000-0001-7248-0662

REVIEWERS' COMMENTS:

Reviewer #1 (Remarks to the Author):

This is a responsive revision which has addressed my main concerns. Two further suggestions:

The authors discuss that they did not discover links between missingness and autistic traits, and caution that this might be due to the population sample. I wonder about the power of this analysis and suggest adding a short comment regarding this aspect.

Minor: "Preregistration: ... as the result OF the review process..."

Reviewer #2 (Remarks to the Author):

Overall, this revised version of the manuscript provides a substantially improved version. Regarding my comments/questions in the first round of review have been addressed thoroughly – to cut a long story short: I am happy with the revised manuscript (bar some rather minor points as noted below) and regard it as a valuable piece of scientific work and a good read for developmental researchers.

I suggest some points, which could be done easily, for further revision to optimize the ms.:

(1) Info/specification of statistical models: On p. 10 (revised ms. marked up pdf – see also your response (#12) to my previous comments), you refer to the model equations in the Supplementary Material – but they are not yet included there (at least not in the respective document which was delivered for this second round of review). Thus, I still would suggest to provide this more clear-cut specification of the statistical model you ran. Also for clarity of reading, regarding the Supplement, Tables S1, S3-S5, S9-11, please provide a short description/clarification (in this Supplement) of the submodels ("Submodel 1" etc.) – e.g., specify/name in the tables' footnotes the restrictions tested with the respective submodel.

(2) On p. 13 (revised ms. marked up pdf): "For the phenotypic association analysis with missing data scores, the first 10 principal components of ancestry were included as covariates" – should I know what are the first 10 principal components of ancestry are? If so, I have to admit embarrassingly that I had to google ... anyhow, what about some kind of hint here (which may also be done by a reference).

(3) On p. 20 (revised ms. marked up pdf): "More advanced methods to deal with missing data provide different solutions by leveraging on the association with observed variables to estimate missing values (e.g., data imputation) and/or distribution parameters (e.g., maximum likelihood strategies)" – I am still a bit unhappy with the wording regarding missing data methods: Couldn't you change "data imputation" to something such as "multiple imputation" (or so), to avoid any notion that single imputation strategies such as mean imputation are a good – or even considerable – missing data method. This comment may appear quite finicky, but my experience is that there are still so many researchers out there who lack in-depth knowledge on modern missing data methods.

Reviewer #3 (Remarks to the Author):

Thank you very much for your careful corrections.

I now have one additional suggestion. The authors have removed the multivariate analysis in the revised version. However, in the Introduction (lines 68–72), they still state that they conducted a multivariate analysis. Please remove these statements from the Introduction. I also recommend eliminating any remaining references to multivariate analysis elsewhere in the manuscript, if applicable.

Dear editor

We thank you and the reviewers very much for all these valuable points and constructive feedback! Please see below how we have addressed the specific points. We hope that the manuscript is now acceptable for publication in Communications Psychology.

We have uploaded one document with all track changes and one clean version.

Best regards,

Terje Falck-Ytter
-on behalf of all authors

The reviewers raised several important concerns. First, we ask that you respond to the concerns by providing a clearer rationale for the selection of the three experimental paradigms (EEG, pupillometry, gaze tracking), including more detail on their representativeness and modality-specific sources of missingness, as well as the sample's relationship to the BATSS study. The introduction should provide a clear justification for the study's hypotheses and explain the proposal why genetic influences violate the MCAR assumption. Please address Reviewer 2's concerns regarding the conceptual interpretation of MCAR, through appropriate additional analyses and a sharpening of the conclusions.

We also ask that you improve the transparency in statistical modeling, and include more information in the main manuscripts, rather than in the SI. With regard to the preregistration, please ensure that your revision is fully in line with the journal's preregistration policy (<https://www.nature.com/commspsychol/editorial-policies/preregistration-policy>). All preregistered analyses should be reported, unless scientifically unfeasible or unsound. Deviations need to be clearly flagged and justified. Please address all methodological concerns, including those pertaining to visual trial exclusions in EEG, handling of composite scores and covariates (e.g., parental education), low power, and multiple testing.

RESPONSE #1: Thanks for summarizing the key issues. Accordingly, we have edited the introduction to provide more clear rationale for the selection of the experiments, and the relationship to the BATSS study. In accordance with R2 we have toned down the interpretations regarding MCAR substantially, and transparently noted the deviations from the preregistration in the beginning of the Methods section.

REVIEWER REPORTS:

Reviewer #1 (Remarks to the Author):

This study examined systematic missingness of data across three infant experiments using a twin-design. A substantial amount of additive genetics and shared environmental effects was

found, which was uncorrelated with additional data collected for characterization of participants (e.g., sex, age, or temperament).

The manuscript is overall very well written, provides important data and a valuable reference to the literature. However, some aspects require clarification in my view.

The intro is extremely short, thereby lacking any specifics regarding missing data in infant studies. While I recognize the need for writing concisely, the problem of missingness in infant studies needs to be characterized based on empirical reports.

RESPONSE #2: Thank you for identifying this issue, we have now rewritten the relevant sentences in the intro and include relevant references to studies of infants using the three methods used in our report.

This also goes for the three experiments selected for this analysis. It should be stated clearly why these were chosen and what, if any, differences between data missingness can be expected. It seems likely that different genetically determined traits contribute to data missingness depending on the method, that is, hair quality and scalp shape might induce bad data quality in EEG, whereas eye physiology might contribute to data loss in eyetracking (EEG: Etienne et al., 2020, Annu. Inf. Conf. IEEE Eng. Med. Biol. Soc., Choi et al., 2021, Affect. Sci; eyetracking: Hessels et al., 2014, Infancy). Moreover, the preparation demands are completely different between EEG and eyetracking, another potential source for differential missingness.

RESPONSE #3: Thank you for this valuable point, we have now integrated these views in our Intro, acknowledging that both shared and unique types of missingness is expected across the experiments. We have also added a sentence motivating the choice of experiments.

Also, in the Results section, the selected experiments are referred to as „representative“. While this term will be debatable in any case, I would agree that eyetracking of looking patterns for human faces is a standard paradigm. However, the EEG and pupillometry experiments run here do not seem representative, but rather highly specific instantiations of these methods. Of course all experiments required a minimum of attention, but attention was not a common DV among experiments. In particular, the experiment referred to as „EEG“ seems to have been an SSVEP experiment, which is not stated clearly. For any novices, this is important information which may also relate to data missingness and thus should be made explicit. The manuscript is not dealing enough with the specifics of the examined experiments and DVs in my view, thus leaving much room for speculation regarding the specificity of familial effects per experiment.

RESPONSE #4: Thank you for this comment. We have now removed all statements claiming our experiments are ‘representative’. We have added a sentence correctly identifying the DV as based on SSVEPs.

The preregistration mentions an exploratory research question, which should also be introduced at the end of the intro („We will also evaluate different multivariate models linking the different missingness indexes, but have no a-priori assumption about what kind of structure will be best fitting due to lack of previous empirical studies.“).

RESPONSE #5: In line with reviewer 3's suggestion, we have removed the multivariate model completely.

While I understand that data preprocessing here is based on the already published primary reports, the selection of trials based on visual inspection in FPVS is not objective and not state of the art. Responses at the stimulated frequency can be quantified as SNR or Z-scores, allowing for an objectified exclusion criterion, which in turn might be related differently to data missingness.

RESPONSE #6: We agree with this point in principle, and we acknowledge this in the discussion as a limitation.

The preregistration described that the medical and psychiatric history information from both parents would be collected. Why was this information not used in the analyses? It might be this background information is related to data missingness, as it might (for example) reflect genetic variance.

Overall, please state very clearly how you deviated from the preregistration and why this was the case.

RESPONSE #7. Here we believe there has been a slight misunderstanding. The preregistration stated that it was a general requirement (inclusion criterion in the larger BATSS study) that "Detailed information about medical and psychiatric history and basic demographic information from both biological parents CAN be obtained, as well as detailed information about the delivery". While we required this for general inclusion, it was not specified that this information would be used in this specific analysis.

The discussion section mentions low power for detecting covariate effects. This should be substantiated by a power analysis.

RESPONSE #8: In accordance with the reviewer3 suggestion we have excluded the multivariate analysis. It is not meaningful to conduct such analyses when the cross-twin-cross-trait correlations are not significant (which we report).

Please specify how parental education was coded for analysis.

RESPONSE #9: Education level was coded on a scale from 1 to 4, where 1 = Primary, 2 = Secondary, 3 = Undergraduate (≤ 3 years) and 4 = Postgraduate level (> 3 years).

Reviewer #2 (Remarks to the Author):

This paper reports results from an analysis of genetic and environmental contributions to data missingness in a Swedish study of infant (5-6 months) twins in widely used 3 experimental paradigms to assess infant brain and behavioral development (EEG, pupillometry and gaze tracking). As I understood, the basic research aim was to evidence that the missing-completely-at-random (MCAR) assumption is typically not valid in this kind of research, as in particular genetic impacts cause missingness, which thus does not occur „randomly“ and statistical

analyses relying on MCAR missing-data treatment may be biased. I have to admit that I am not an expert in infant research, hence cannot say whether it indeed suffers from unreflected and unjustifiable use of MCAR methodology, but if so, it would be an important methodological topic for this field of research. However, I have a serious concern regarding the authors' conceptual understanding of MCAR (and the respective theory of "missingness mechanisms" grounded on Rubin's [1979] seminal work):

(1) Major point: IMO, this study's general "rationale" is affected by a quite common misunderstanding of MCAR (as well as MAR and MNAR), considering these concepts as kind of causal theory of data missingness (see, e.g., Schafer & Graham, 2002, p. 151: "... Rubin's (1976) definitions describe statistical relationships between the data and the missingness, not causal relationships"). That is, MCAR etc. are stochastic concepts, regarding the associations between the scores/values and the missingness of the variables analyzed with a statistical model. Thus, neither does knowledge of a cause of missingness in the assessments of a variable strictly imply that it cannot be MCAR whenever the variable is used in any statistical analysis, nor is knowledge of such causes needed to prove the MCAR does not hold. Given a set of variables (i.e., a multivariate distribution to be analyzed with a statistical model), MCAR means statistical independence of a variable's missingness from all these variables (i.e., the observed and unobserved scores). Therefore, having evidenced that interindividual variation of missingness is partly due to genetic variation, the conclusion that "we can generally reject the common (explicit or implicit) assumption of data being missing completely at random" (page 8, line 177-179) is not warranted offhandedly. What would be needed in addition is some evidence that indeed the same (or similar) genetic impacts that increase the missingness probability do also affect the scores in the respective variables or at least some other variables included in a statistical analysis model, hence causing statistical associations between the missingness and the data. The Rubin concepts of missingness mechanisms imply, that these are specific to the model/set of variables analyzed (e.g., adding variables predictive of another variable's missingness may shift the latter from MNAR to MAR).

Thus, I strongly recommend (1) to "downtone" and prevent the notion of causality when referring to concepts such as MCAR (which seems particularly important as this causal misconception indeed is widespread among applied researchers), and (2) to strengthen and clarify the relevant insights to be gained from the analyses/results reported – my above critique did not mean to say that the disclosure of genetic variance "causing" missingness is not a relevant scientific goal here, I just doubt that this disclosure is generally relevant with respect to the application of MCAR vs. MAR or MNAR methods for the analysis of these measures. Moreover, hasn't it by now long been accepted that at least MAR missing data treatment is always preferable to MCAR in presence of substantial shares of missing data – would researchers really need evidence of genetic impacts on missingness (even if this would be relevant with regard to MCAR) to do so?

RESPONSE #10: Thank you for this valid point, which has led us to indeed substantially tone down the MCAR argumentation linked to missingness in the paper. This change applies to the abstract, introduction and discussion, and deviations from preregistration are noted in the Methods.

(2) Given the rather general specification of a hypothetical prediction to be tested in this study (p. 3-4, l. 83-87), I wondered about the multiple testing involved in the analyses (nested model

comparisons, tests of trial level missingness): What is/are the familywise error rate/s here, potentially needing some adjustments for multiple testing? If the study hypothesis indeed is just that some genetic effect will be found on some of the statistical tests, multiple testing adjustments would of course be needed. Thus, the study hypotheses should be stated more exactly and specifically. Also, as explained in my above point (1), I doubt that the prediction made on the bottom of p.3 is “contrary to the MCAR hypothesis”.

RESPONSE #11: Thank you for raising an important point. In twin modelling, model comparisons such as ACE vs AE vs CE are conducted within a structural equation modelling framework and are considered part of the model fitting process rather than independent hypothesis tests. These are not usually subject to family-wise error correction in the classical sense.

While we did test missing data models across multiple tasks, we did not conduct classical hypothesis tests as model selection was largely based on AIC/BIC criteria. To guard against overfitting, we restricted our interpretation to models showing both (a) a best-fitting AE or CE structure and (b) confidence intervals excluding zero for the relevant variance components. We also report full model fit indices and estimates for all traits in the Supplementary Material. However, as the reviewer correctly points out, we subsequently conducted likelihood ratio tests (LRTs) to assess the significance of A and C components across four models (one for each of the three experiments and one composite score). These tests yielded a total of 8 comparisons (2 variance components × 4 models) for the hypotheses tested on experiment-level data missingness, and 6 comparisons (2 variance components × 3 models) on trial-level data missingness. To address this, we have now applied a conservative Bonferroni correction for multiple comparisons and updated results accordingly.

(3) I found it difficult to follow your analytical strategy without a more detailed and clear-cut presentation of the statistical models. Of course, readers of this manuscript may/should already know what a liability threshold model is and/or may gain such knowledge from reading respective methodological literature, but I doubt that this method is as well known as the most widely used regression models. It would facilitate reading if one could just somehow see the exact model used here – e.g. the model equations including the variables/measures modelled. This may, however, already have been presented in the Supplementary Material – I couldn't open some of these pdf-files.

RESPONSE #12: Thank you for the suggestion. We understand that the analytic pipeline might not be clear to follow, we have tried to clarify the core steps in the Methods. To improve clarity, we have also added the model equations to the Supplementary Material, as requested.

(4) Minor point, related to (3): “All twin modelling assumptions were met” (p. 5, l.101) – appeared to me a bit confusing, as one may think of modelling assumptions such as multivariate normality of the joint twin pair liability distribution (e.g. Benckek & Morris, 2013, Hum. Genet., doi: 10.1007/s00439-013-1334-z – if this is the analytic model you used). BTW, this latter study suggests that the ACE liability threshold model is highly vulnerable to violations of the MVN assumption – maybe also worth some consideration for your study?

RESPONSE #13: We agree with the reviewer that the statement could be misleading. We have therefore clarified that twin modelling assumptions of equality of mean and variances across twin order and zygosity were met.

(5) Minor point: On bottom of p. 2 (l. 57) and also later in the manuscript when you mention “individual variability”, I wondered whether you mean inter- or intra-individual variability (or both). Maybe some clarification here?

RESPONSE #14: We have clarified whether it is intra- or inter- individual variability, or both, throughout the manuscript.

(6) P. 21: “We chose to include reasons of missingness recorded by the research assistants as ‘technical reasons’ because ... in the data” – I couldn’t understand what this means to say. E.g.: You call something technical reason “because it is difficult to know for sure whether a ‘technical reason’ is not at all linked to the child”? Could this be explained more clear-cut? I may, however, just have been slow on the uptake here.

RESPONSE #15: We agree with the reviewer that the phrasing was unclear so we clarified in the Methods. What we meant by that is that we chose to include experimental events coded as missing based on technical reasons anyway in our analysis, instead of excluding those participants.

Reviewer #3 (Remarks to the Author):

The paper investigated the genetic and environmental etiology of data missingness in infant twins and found that, while experiment-level missingness was influenced by shared and non-shared environmental factors (with the exception of EEG), trial-level missingness was influenced by genetic and non-shared environmental factors. Given that the etiology of data missingness has been rarely studied, the results provide a valuable contribution to the literature on this topic. Below are my comments and suggestions for improvement:

1. Introduction: Apparently, The BATSS dataset includes a wide range of data from infant twins. However, it is unclear why the authors specifically chose three variables—EEG, pupillometry, and gaze tracking—to study the etiology of data missingness. It is possible that other infant data may exhibit different etiological patterns. While the reason for this selection is briefly mentioned in the Discussion section, I suggest providing a more detailed and clear explanation for why these three variables were chosen in the Introduction.

RESPONSE #16: Thank you for this suggestion, which is in line also with reviewer1; we have now clarified this better in the intro.

2. Introduction: In the last paragraph, the authors hypothesized that genetic and shared environmental influences would affect the missing data, contrary to the assumption of missing

completely at random (MCAR). However, the rationale for this hypothesis is not clearly explained in the Introduction. Providing a justification for this hypothesis in the Introduction would strengthen the paper and give readers a better understanding of the basis for this assumption.

RESPONSE #17: in light of the points by reviewer2, we have opted to substantially tone down the specific implications for MCAR in the intro and the paper generally. This constitutes a deviation from the preregistration, which we acknowledge explicitly in the revised paper.

3. Title: The term "Developmental Psychology" is included in the title of the paper. However, Developmental Psychology is a broad discipline that encompasses various subfields, including epidemiological research and studies on older adults. Given the focus of this study, I am concerned that the results may not be easily generalized to the field of Developmental Psychology as a whole. It may be more appropriate to specify a more focused research area, such as neurodevelopmental infant research, in the title.

RESPONSE #18: Thanks for this suggestion, we have updated the title accordingly.
(developmental cognitive neuroscience)

4. Line 72- unique environmental influences include "measurement error". Please specify this and explain what "measurement error" means in this particular study.

RESPONSE #19: Good suggestion, we have added an interpretation of measurement error in the context of this study in the Discussion.

5. Sample- While the reference to the sample and the inclusion and exclusion criteria have been provided, there seems to be a lack of detailed information regarding the recruitment process. It would be helpful for readers to understand how the authors recruited participants for the study. Specifically, were participants recruited from a larger database, or was the sample selected from a particular subset of the population? Providing details on the recruitment process would offer transparency and allow readers to better assess the generalizability of the findings.

RESPONSE #20: This has now been added to the Methods (participant) section.

6. Additionally, it might be useful to state the general aim of the BATSS (presumably the larger study or dataset from which the sample was drawn) to give readers context for the sample's characteristics.

RESPONSE #21: Thanks for this suggestion; now added to the Methods (participant) section

7. It would be important to clarify whether the current sample includes all twins within the BATSS dataset or only those who participated in the specific experiments referenced. Understanding whether the sample is representative of the entire BATSS cohort or restricted to those who volunteered for certain experiments would help clarify the scope of the findings and their applicability.

RESPONSE #22: The sample analysed here reflects the whole BATSS sample, with some minor exception (infants who were included in BATSS and tested, but upon closer inspection were found not to fulfill general BATSS criteria, most frequently presence of twin-to-twin transfusion syndrome. Concretely: 622 infants were included in BATSS, but 594 (ie approx 95%) were included in this analysis. This information is described towards the end of the Methods (participants) section.

8. Table 1: The number of missing data is consistently higher, and the number of valid trials is consistently lower in MZ twins compared to DZ twins, although these differences did not always reach statistical significance. Nevertheless, the consistent patterns are intriguing. Do the authors have any speculation for these patterns?

RESPONSE #23: We agree this is an interesting observation. Partly it could reflect the trivial fact that there are more MZ. But it could also be some real effects, possibly subtle zygosity-related factors influencing data quality or task engagement. Or attention span or temperament.

9. Please state how you calculated the composite score (Table 2).

RESPONSE #24: The composite score was operationalised as a binary indicator of missingness across the three experimental paradigms. Specifically, the composite score was coded as 1 if an infant had missing data in one or more of the three experiments, and as 0 if data were available for all three.

10. The three variables are not highly correlated, and the sample size is relatively small. As a result, the study is underpowered to conduct a multivariate analysis. The multivariate analysis section introduces some confusion and unnecessary complexity. I suggest that the authors remove the multivariate twin analysis from both the manuscript and the supplementary information, and instead report the correlations among the three variables with discussion in the Results or Discussion section to avoid confusion.

RESPONSE #25: We have now removed the Multivariate analysis, and provided a motivation for this deviation in at the start of the Methods section.

Dear editor

We thank you and the reviewers very much for all these valuable points and constructive feedback! Please see below how we have addressed the specific points. We hope that the manuscript is now acceptable for publication in Communications Psychology.

We have uploaded one document with all track changes and one clean version.

Best regards,

Terje Falck-Ytter
-on behalf of all authors

The reviewers raised several important concerns. First, we ask that you respond to the concerns by providing a clearer rationale for the selection of the three experimental paradigms (EEG, pupillometry, gaze tracking), including more detail on their representativeness and modality-specific sources of missingness, as well as the sample's relationship to the BATSS study. The introduction should provide a clear justification for the study's hypotheses and explain the proposal why genetic influences violate the MCAR assumption. Please address Reviewer 2's concerns regarding the conceptual interpretation of MCAR, through appropriate additional analyses and a sharpening of the conclusions.

We also ask that you improve the transparency in statistical modeling, and include more information in the main manuscripts, rather than in the SI. With regard to the preregistration, please ensure that your revision is fully in line with the journal's preregistration policy (<https://www.nature.com/commspsychol/editorial-policies/preregistration-policy>). All preregistered analyses should be reported, unless scientifically unfeasible or unsound. Deviations need to be clearly flagged and justified. Please address all methodological concerns, including those pertaining to visual trial exclusions in EEG, handling of composite scores and covariates (e.g., parental education), low power, and multiple testing.

RESPONSE #1: Thanks for summarizing the key issues. Accordingly, we have edited the introduction to provide more clear rationale for the selection of the experiments, and the relationship to the BATSS study. In accordance with R2 we have toned down the interpretations regarding MCAR substantially, and transparently noted the deviations from the preregistration in the beginning of the Methods section.

REVIEWER REPORTS:

Reviewer #1 (Remarks to the Author):

This study examined systematic missingness of data across three infant experiments using a twin-design. A substantial amount of additive genetics and shared environmental effects was

found, which was uncorrelated with additional data collected for characterization of participants (e.g., sex, age, or temperament).

The manuscript is overall very well written, provides important data and a valuable reference to the literature. However, some aspects require clarification in my view.

The intro is extremely short, thereby lacking any specifics regarding missing data in infant studies. While I recognize the need for writing concisely, the problem of missingness in infant studies needs to be characterized based on empirical reports.

RESPONSE #2: Thank you for identifying this issue, we have now rewritten the relevant sentences in the intro and include relevant references to studies of infants using the three methods used in our report.

This also goes for the three experiments selected for this analysis. It should be stated clearly why these were chosen and what, if any, differences between data missingness can be expected. It seems likely that different genetically determined traits contribute to data missingness depending on the method, that is, hair quality and scalp shape might induce bad data quality in EEG, whereas eye physiology might contribute to data loss in eyetracking (EEG: Etienne et al., 2020, Annu. Inf. Conf. IEEE Eng. Med. Biol. Soc., Choi et al., 2021, Affect. Sci; eyetracking: Hessels et al., 2014, Infancy). Moreover, the preparation demands are completely different between EEG and eyetracking, another potential source for differential missingness.

RESPONSE #3: Thank you for this valuable point, we have now integrated these views in our Intro, acknowledging that both shared and unique types of missingness is expected across the experiments. We have also added a sentence motivating the choice of experiments.

Also, in the Results section, the selected experiments are referred to as „representative“. While this term will be debatable in any case, I would agree that eyetracking of looking patterns for human faces is a standard paradigm. However, the EEG and pupillometry experiments run here do not seem representative, but rather highly specific instantiations of these methods. Of course all experiments required a minimum of attention, but attention was not a common DV among experiments. In particular, the experiment referred to as „EEG“ seems to have been an SSVEP experiment, which is not stated clearly. For any novices, this is important information which may also relate to data missingness and thus should be made explicit. The manuscript is not dealing enough with the specifics of the examined experiments and DVs in my view, thus leaving much room for speculation regarding the specificity of familial effects per experiment.

RESPONSE #4: Thank you for this comment. We have now removed all statements claiming our experiments are ‘representative’. We have added a sentence correctly identifying the DV as based on SSVEPs.

The preregistration mentions an exploratory research question, which should also be introduced at the end of the intro („We will also evaluate different multivariate models linking the different missingness indexes, but have no a-priori assumption about what kind of structure will be best fitting due to lack of previous empirical studies.“).

RESPONSE #5: In line with reviewer 3's suggestion, we have removed the multivariate model completely.

While I understand that data preprocessing here is based on the already published primary reports, the selection of trials based on visual inspection in FPVS is not objective and not state of the art. Responses at the stimulated frequency can be quantified as SNR or Z-scores, allowing for an objectified exclusion criterion, which in turn might be related differently to data missingness.

RESPONSE #6: Thanks for pointing out this important point. We have revised the manuscript to make clearer that trial-level missingness (N cycles) was based on voltage range exceeding 100uV. Thus, this part was fully automatic and standardized. Experiment level missingness was based on visual inspection of activity in the visual cortex at the stimulated frequency (in the local Form condition, stimulus duration 2 min), using all available data from each participant. In light of the reviewer's concern, we performed an inter-rater reliability check of these visual-inspection-based decisions, finding substantial to near perfect agreement (see Methods (EEG experiment) and Limitations). Indeed, as the reviewer writes, these were the rules used in the original studies.

The preregistration described that the medical and psychiatric history information from both parents would be collected. Why was this information not used in the analyses? It might be this background information is related to data missingness, as it might (for example) reflect genetic variance.

Overall, please state very clearly how you deviated from the preregistration and why this was the case.

RESPONSE #7. Here we believe there has been a slight misunderstanding. The preregistration stated that it was a general requirement (inclusion criterion in the larger BATSS study) that "Detailed information about medical and psychiatric history and basic demographic information from both biological parents CAN be obtained, as well as detailed information about the delivery". While we required this for general inclusion, it was not specified that this information would be used in this specific analysis.

The discussion section mentions low power for detecting covariate effects. This should be substantiated by a power analysis.

RESPONSE #8: In accordance with the reviewer3 suggestion we have excluded the multivariate analysis. It is not meaningful to conduct such analyses when the cross-twin-cross-trait correlations are not significant (which we report).

Please specify how parental education was coded for analysis.

RESPONSE #9: Education level was coded on a scale from 1 to 4, where 1 = Primary, 2 = Secondary, 3 = Undergraduate (≤ 3 years) and 4 = Postgraduate level (> 3 years).

Reviewer #2 (Remarks to the Author):

This paper reports results from an analysis of genetic and environmental contributions to data missingness in a Swedish study of infant (5-6 months) twins in widely used 3 experimental paradigms to assess infant brain and behavioral development (EEG, pupillometry and gaze tracking). As I understood, the basic research aim was to evidence that the missing-completely-at-random (MCAR) assumption is typically not valid in this kind of research, as in particular genetic impacts cause missingness, which thus does not occur „randomly“ and statistical analyses relying on MCAR missing-data treatment may be biased. I have to admit that I am not an expert in infant research, hence cannot say whether it indeed suffers from unreflected and unjustifiable use of MCAR methodology, but if so, it would be an important methodological topic for this field of research. However, I have a serious concern regarding the authors' conceptual understanding of MCAR (and the respective theory of “missingness mechanisms” grounded on Rubin’s [1979] seminal work):

(1) Major point: IMO, this study’s general “rationale” is affected by a quite common misunderstanding of MCAR (as well as MAR and MNAR), considering these concepts as kind of causal theory of data missingness (see, e.g., Schafer & Graham, 2002, p. 151: “... Rubin’s (1976) definitions describe statistical relationships between the data and the missingness, not causal relationships”). That is, MCAR etc. are stochastic concepts, regarding the associations between the scores/values and the missingness of the variables analyzed with a statistical model. Thus, neither does knowledge of a cause of missingness in the assessments of a variable strictly imply that it cannot be MCAR whenever the variable is used in any statistical analysis, nor is knowledge of such causes needed to prove the MCAR does not hold. Given a set of variables (i.e., a multivariate distribution to be analyzed with a statistical model), MCAR means statistical independence of a variable’s missingness from all these variables (i.e., the observed and unobserved scores). Therefore, having evidenced that interindividual variation of missingness is partly due to genetic variation, the conclusion that “we can generally reject the common (explicit or implicit) assumption of data being missing completely at random” (page 8, line 177-179) is not warranted offhandedly. What would be needed in addition is some evidence that indeed the same (or similar) genetic impacts that increase the missingness probability do also affect the scores in the respective variables or at least some other variables included in a statistical analysis model, hence causing statistical associations between the missingness and the data. The Rubin concepts of missingness mechanisms imply, that these are specific to the model/set of variables analyzed (e.g., adding variables predictive of another variable’s missingness may shift the latter from MNAR to MAR).

Thus, I strongly recommend (1) to “downtone” and prevent the notion of causality when referring to concepts such as MCAR (which seems particularly important as this causal misconception indeed is widespread among applied researchers), and (2) to strengthen and clarify the relevant insights to be gained from the analyses/results reported – my above critique did not mean to say that the disclosure of genetic variance “causing” missingness is not a relevant scientific goal here, I just doubt that this disclosure is generally relevant with respect to the application of MCAR vs. MAR or MNAR methods for the analysis of these measures. Moreover, hasn’t it by now long been accepted that at least MAR missing data treatment is always preferable to MCAR in presence of substantial shares of missing data – would researchers really need evidence of genetic impacts on missingness (even if this would be relevant with regard to MCAR) to do so?

RESPONSE #10: Thank you for this valid point, which has led us to indeed substantially tone down the MCAR argumentation linked to missingness in the paper. This change applies to the abstract, introduction and discussion, and deviations from preregistration are noted in the Methods.

(2) Given the rather general specification of a hypothetical prediction to be tested in this study (p. 3-4, l. 83-87), I wondered about the multiple testing involved in the analyses (nested model comparisons, tests of trial level missingness): What is/are the familywise error rate/s here, potentially needing some adjustments for multiple testing? If the study hypothesis indeed is just that some genetic effect will be found on some of the statistical tests, multiple testing adjustments would of course be needed. Thus, the study hypotheses should be stated more exactly and specifically. Also, as explained in my above point (1), I doubt that the prediction made on the bottom of p.3 is “contrary to the MCAR hypothesis”.

RESPONSE #11: Thank you for raising an important point. In twin modelling, model comparisons such as ACE vs AE vs CE are conducted within a structural equation modelling framework and are considered part of the model fitting process rather than independent hypothesis tests. These are not usually subject to family-wise error correction in the classical sense.

While we did test missing data models across multiple tasks, we did not conduct classical hypothesis tests as model selection was largely based on AIC/BIC criteria. To guard against overfitting, we restricted our interpretation to models showing both (a) a best-fitting AE or CE structure and (b) confidence intervals excluding zero for the relevant variance components. We also report full model fit indices and estimates for all traits in the Supplementary Material. However, as the reviewer correctly points out, we subsequently conducted likelihood ratio tests (LRTs) to assess the significance of A and C components across four models (one for each of the three experiments and one composite score). These tests yielded a total of 8 comparisons (2 variance components × 4 models) for the hypotheses tested on experiment-level data missingness, and 6 comparisons (2 variance components × 3 models) on trial-level data missingness. To address this, we have now applied a conservative Bonferroni correction for multiple comparisons and updated results accordingly.

(3) I found it difficult to follow your analytical strategy without a more detailed and clear-cut presentation of the statistical models. Of course, readers of this manuscript may/should already know what a liability threshold model is and/or may gain such knowledge from reading respective methodological literature, but I doubt that this method is as well known as the most widely used regression models. It would facilitate reading if one could just somehow see the exact model used here – e.g. the model equations including the variables/measures modelled. This may, however, already have been presented in the Supplementary Material – I couldn’t open some of these pdf-files.

RESPONSE #12: Thank you for the suggestion. We understand that the analytic pipeline might not be clear to follow, we have tried to clarify the core steps in the Methods. To improve clarity, we have also added the model equations to the Supplementary Material, as requested.

(4) Minor point, related to (3): “All twin modelling assumptions were met” (p. 5, l.101) – appeared to me a bit confusing, as one may think of modelling assumptions such as multivariate

normality of the joint twin pair liability distribution (e.g. Benckek & Morris, 2013, Hum. Genet., doi: 10.1007/s00439-013-1334-z – if this is the analytic model you used). BTW, this latter study suggests that the ACE liability threshold model is highly vulnerable to violations of the MVN assumption – maybe also worth some consideration for your study?

RESPONSE #13: We agree with the reviewer that the statement could be misleading. We have therefore clarified that twin modelling assumptions of equality of mean and variances across twin order and zygosity were met.

(5) Minor point: On bottom of p. 2 (l. 57) and also later in the manuscript when you mention “individual variability”, I wondered whether you mean inter- or intra-individual variability (or both). Maybe some clarification here?

RESPONSE #14: We have clarified whether it is intra- or inter- individual variability, or both, throughout the manuscript.

(6) P. 21: “We chose to include reasons of missingness recorded by the research assistants as ‘technical reasons’ because ... in the data” – I couldn’t understand what this means to say. E.g.: You call something technical reason “because it is difficult to know for sure whether a ‘technical reason’ is not at all linked to the child”? Could this be explained more clear-cut? I may, however, just have been slow on the uptake here.

RESPONSE #15: We agree with the reviewer that the phrasing was unclear so we clarified in the Methods. What we meant by that is that we chose to include experimental events coded as missing based on technical reasons anyway in our analysis, instead of excluding those participants.

Reviewer #3 (Remarks to the Author):

The paper investigated the genetic and environmental etiology of data missingness in infant twins and found that, while experiment-level missingness was influenced by shared and non-shared environmental factors (with the exception of EEG), trial-level missingness was influenced by genetic and non-shared environmental factors. Given that the etiology of data missingness has been rarely studied, the results provide a valuable contribution to the literature on this topic. Below are my comments and suggestions for improvement:

1. Introduction: Apparently, The BATSS dataset includes a wide range of data from infant twins. However, it is unclear why the authors specifically chose three variables—EEG, pupillometry, and gaze tracking—to study the etiology of data missingness. It is possible that other infant data may exhibit different etiological patterns. While the reason for this selection is briefly mentioned in the Discussion section, I suggest providing a more detailed and clear explanation for why these three variables were chosen in the Introduction.

RESPONSE #16: Thank you for this suggestion, which is in line also with reviewer1; we have now clarified this better in the intro.

2. Introduction: In the last paragraph, the authors hypothesized that genetic and shared environmental influences would affect the missing data, contrary to the assumption of missing completely at random (MCAR). However, the rationale for this hypothesis is not clearly explained in the Introduction. Providing a justification for this hypothesis in the Introduction would strengthen the paper and give readers a better understanding of the basis for this assumption.

RESPONSE #17: in light of the points by reviewer2, we have opted to substantially tone down the specific implications for MCAR in the intro and the paper generally. This constitutes a deviation from the preregistration, which we acknowledge explicitly in the revised paper.

3. Title: The term "Developmental Psychology" is included in the title of the paper. However, Developmental Psychology is a broad discipline that encompasses various subfields, including epidemiological research and studies on older adults. Given the focus of this study, I am concerned that the results may not be easily generalized to the field of Developmental Psychology as a whole. It may be more appropriate to specify a more focused research area, such as neurodevelopmental infant research, in the title.

RESPONSE #18: Thanks for this suggestion, we have updated the title accordingly.
(developmental cognitive neuroscience)

4. Line 72- unique environmental influences include "measurement error". Please specify this and explain what "measurement error" means in this particular study.

RESPONSE #19: Good suggestion, we have added an interpretation of measurement error in the context of this study in the Discussion.

5. Sample- While the reference to the sample and the inclusion and exclusion criteria have been provided, there seems to be a lack of detailed information regarding the recruitment process. It would be helpful for readers to understand how the authors recruited participants for the study. Specifically, were participants recruited from a larger database, or was the sample selected from a particular subset of the population? Providing details on the recruitment process would offer transparency and allow readers to better assess the generalizability of the findings.

RESPONSE #20: This has now been added to the Methods (participant) section.

6. Additionally, it might be useful to state the general aim of the BATSS (presumably the larger study or dataset from which the sample was drawn) to give readers context for the sample's characteristics.

RESPONSE #21: Thanks for this suggestion; now added to the Methods (participant) section

7. It would be important to clarify whether the current sample includes all twins within the BATSS dataset or only those who participated in the specific experiments referenced.

Understanding whether the sample is representative of the entire BATSS cohort or restricted to those who volunteered for certain experiments would help clarify the scope of the findings and their applicability.

RESPONSE #22: The sample analysed here reflects the whole BATSS sample, with some minor exception (infants who were included in BATSS and tested, but upon closer inspection were found not to fulfill general BATSS criteria, most frequently presence of twin-to-twin transfusion syndrome. Concretely: 622 infants were included in BATSS, but 594 (ie approx 95%) were included in this analysis. This information is described towards the end of the Methods (participants) section.

8. Table 1: The number of missing data is consistently higher, and the number of valid trials is consistently lower in MZ twins compared to DZ twins, although these differences did not always reach statistical significance. Nevertheless, the consistent patterns are intriguing. Do the authors have any speculation for these patterns?

RESPONSE #23: We agree this is an interesting observation. Partly it could reflect the trivial fact that there are more MZ. But it could also be some real effects, possibly subtle zygoty-related factors influencing data quality or task engagement. Or attention span or temperament.

9. Please state how you calculated the composite score (Table 2).

RESPONSE #24: The composite score was operationalised as a binary indicator of missingness across the three experimental paradigms. Specifically, the composite score was coded as 1 if an infant had missing data in one or more of the three experiments, and as 0 if data were available for all three.

10. The three variables are not highly correlated, and the sample size is relatively small. As a result, the study is underpowered to conduct a multivariate analysis. The multivariate analysis section introduces some confusion and unnecessary complexity. I suggest that the authors remove the multivariate twin analysis from both the manuscript and the supplementary information, and instead report the correlations among the three variables with discussion in the Results or Discussion section to avoid confusion.

RESPONSE #25: We have now removed the Multivariate analysis, and provided a motivation for this deviation in at the start of the Methods section.

Dear Editor,

Thanks to you and the reviewer for further comments, which we have addressed below, in the Editorial Request Table, and in the manuscript itself.

We hope to have addressed all the remaining issues.

Best

Terje Falck-Ytter & Giorgia Bussu

Reviewer #1 (Remarks to the Author):

This is a responsive revision which has addressed my main concerns. Two further suggestions:

The authors discuss that they did not discover links between missingness and autistic traits, and caution that this might be due to the population sample. I wonder about the power of this analysis and suggest adding a short comment regarding this aspect.

RESPONSE #1: We thank the reviewer for raising this important point. While our sample is relatively large for infant twin research ($n = 594$), power to detect small associations may be limited. We have now added a brief note in the *Discussion* acknowledging that the absence of significant associations may reflect limited power, especially given the modest effect sizes typically observed in this domain and the use of stringent correction procedures.

Minor: "Preregistration: ... as the result OF the review process..."

RESPONSE #2: thanks for catching this – changed.

Reviewer #2 (Remarks to the Author):

Overall, this revised version of the manuscript provides a substantially improved version. Regarding my comments/questions in the first round of review have been addressed thoroughly – to cut a long story short: I am happy with the revised manuscript (bar some rather minor points as noted below) and regard it as a valuable piece of scientific work and a good read for developmental researchers.

I suggest some points, which could be done easily, for further revision to optimize the ms.:

(1) Info/specification of statistical models: On p. 10 (revised ms. marked up pdf – see also your response (#12) to my previous comments), you refer to the model equations in the Supplementary Material – but they are not yet included there (at least not in the respective document which was delivered for this second round of review). Thus, I still would suggest to provide this more clear-cut specification of the statistical model you ran. Also for clarity of reading, regarding the Supplement, Tables S1, S3-S5, S9-11, please provide a short description/clarification (in this Supplement) of the submodels (“Submodel 1” etc.) – e.g., specify/name in the tables’ footnotes the restrictions tested with the respective submodel.

RESPONSE #3: We thank the reviewer for this helpful comment. We have now added a specification of the statistical models in the *Supplementary Material (Supplementary Methods 1)*, including the equations underlying the liability-threshold and ACE models. In addition, we have clarified the meaning of the nested submodels tested (e.g., equality constraints on means/variances across twin order and zygosity, and parameter constraints in AE/CE/E models) with a brief description of each submodel in the table footnotes.

(2) On p. 13 (revised ms. marked up pdf): “For the phenotypic association analysis with missing data scores, the first 10 principal components of ancestry were included as covariates” – should I know what are the first 10 principal components of ancestry are? If so, I have to admit embarrassingly that I had to google ... anyhow, what about some kind of hint here (which may also be done by a reference).

RESPONSE #4: We agree that this statement may not be immediately clear to all readers. We have now added a brief clarification explaining that principal components of ancestry are commonly used to control for population stratification in genetic analyses.

(3) On p. 20 (revised ms. marked up pdf): “More advanced methods to deal with missing data provide different solutions by leveraging on the association with observed variables to estimate missing values (e.g., data imputation) and/or distribution parameters (e.g., maximum likelihood strategies)” – I am still a bit unhappy with the wording regarding missing data methods: Couldn’t you change “data imputation” to something such as “multiple imputation” (or so), to avoid any notion that single imputation strategies such as mean imputation are a good – or even considerable – missing data method. This comment may appear quite finicky, but my experience is that there are still so many researchers out there who lack in-depth knowledge on modern missing data methods.

RESPONSE #5: We agree with the reviewer and have revised the wording to explicitly refer to modern approaches such as multiple imputation, to avoid any ambiguity regarding outdated single imputation methods.

Reviewer #3 (Remarks to the Author):

Thank you very much for your careful corrections.

I now have one additional suggestion. The authors have removed the multivariate analysis in the revised version. However, in the Introduction (lines 68–72), they still state that they conducted a multivariate analysis. Please remove these statements from the Introduction. I also recommend eliminating any remaining references to multivariate analysis elsewhere in the manuscript, if applicable.

RESPONSE #6: thanks for noting this – now removed.